# ONLINE LOW-RANK APPROXIMATION VIA ADAPTIVE SPHERICAL PARTITIONING

## ABSTRACT

We study the problem of online low-rank approximation, where at each time step an algorithm receives a new vector and must maintain a rank-$k$ subspace that serves as a compressed representation of the data. The specific formulation we use is the weighted low-rank approximation (WLRA) objective: at each step, the algorithm incurs loss equal to the weighted squared reconstruction error of the incoming point with respect to its current subspace. The goal is to minimize regret against the best rank-$k$ subspace in hindsight, whose reconstruction cost we denote by $\mathcal{C}$. We first establish an online-to-offline reduction: the existence of an efficient no-regret online algorithm for WLRA would imply an efficient approximation scheme for the offline problem, which is unlikely under standard complexity assumptions. Although WLRA is APX-hard in the offline setting, we show that the standard Multiplicative Weights Update Algorithm (MWUA) can achieve sublinear regret in expectation with respect to a $(1 + \varepsilon)$-multiplicative approximation of $\mathcal{C}$. Specifically, we use an adaptive spherical hierarchical region decomposition that iteratively refines the $d$-dimensional unit sphere $\mathbb{S}^d$ based on the density of the data. At each split, a region is partitioned into $2^{d-1}$ sub-regions, producing a hierarchal tree decomposition, while our algorithm maintains centroids of the points in each region as the set of experts. Finally, we complement our theoretical results with empirical evaluations that demonstrate the efficiency of our algorithm compared to previous baselines.

## 1    INTRODUCTION

Modern applications generate data at massive scales, driven by the shift of everyday activities to digital platforms. Processing such data streams efficiently is a central challenge in machine learning, as conventional algorithms often become computationally prohibitive. Introduced by Eckart & Young (1936), low-rank approximation has long been a cornerstone of data analysis, providing a unifying framework for compressing large datasets, uncovering latent structure, and improving computational efficiency. Formally, given a data matrix $\mathbf{A} \in \mathbb{R}^{n \times d}$, the goal is to approximate $\mathbf{A}$ by a product of low-rank factors $\mathbf{U} \in \mathbb{R}^{n \times k}$ and $\mathbf{V} \in \mathbb{R}^{k \times d}$ that minimize the Frobenius norm loss $\|\mathbf{A} - \mathbf{UV}\|_F^2$. This formulation reduces dimensionality while retaining the most informative components, enabling interpretability and efficiency. These advantages have made low-rank methods pervasive in machine learning, powering applications from recommendation systems, e.g., the Netflix Prize (Bell et al., 2007; 2008; Koren, 2009; Bell et al., 2010), to modern foundation models, where techniques such as LoRA (Hu et al., 2022; Xu et al., 2024; Wu et al., 2024; Li et al., 2024) enable efficient fine-tuning by factoring weight updates into low-rank components. By approximating $\mathbf{A}$ with only $(n + d)k$ parameters rather than $nd$, low-rank approximation dramatically reduces storage and speeds up matrix-vector multiplication, which is critical in large-scale settings.

In this work, we study *online* low-rank approximation, where rows of $\mathbf{A}$ arrive sequentially, and at each time $t$, the algorithm must select a rank-$k$ basis $\mathbf{V}_t$ *before* seeing $x_t$, incurring loss

$$\ell(\mathbf{V}_t, x_t) = \|x_t - \mathbf{V}_t \mathbf{V}_t^\top x_t\|_2^2.$$

This streaming formulation naturally arises in recommendation systems, fraud detection (Kamp & Boley, 2019), and adaptive learning pipelines, where the full dataset is unknown in advance and recomputing the SVD at each step is computationally infeasible. Performance is measured using the

*regret* of the algorithm over $T$ time steps: the difference between the cumulative projection loss of the online algorithm and that of the optimal rank-$k$ subspace chosen in hindsight:

$$\text{Regret}_T = \sum_{t=1}^{T} \ell(\mathbf{V}_t, x_t) - \min_{\mathbf{V}^* \in \mathbb{R}^{d \times k}} \sum_{t=1}^{T} \|x_t - \mathbf{V}^*(\mathbf{V}^*)^\top x_t\|_2^2.$$

Our goal is to design online algorithms with *sublinear regret*, meaning that the average loss per time step approaches that of the optimal offline solution as $T$ grows.

This online low-rank approximation problem can be viewed through the lens of classical *online learning* frameworks, particularly *learning with experts*. In this perspective, each candidate rank-$k$ basis can be treated as an *expert*, and the algorithm must select a basis before observing the incoming vector, analogous to choosing an expert before seeing the outcome. The incurred loss corresponds to the residual projection error, similar to the payoff (or loss) of the chosen expert. Regret minimization in this context is directly analogous to minimizing cumulative loss relative to the best fixed expert in hindsight. This connection seems to allow techniques from online learning, such as the Multiplicative Weights Update Algorithm (MWUA) (Arora et al., 2012), to be adapted to the low-rank setting, potentially enabling algorithms that adaptively combine candidate bases and achieve provably low regret. Unfortunately, such algorithms provably incur regret $\Omega(\sqrt{T \log n})$, where $n$ is the number of experts (Gravin et al., 2017), which corresponds to the number of possible rank-$k$ bases in our setting. Naïvely, the number of such bases could be infinite and even more involved techniques such as preemptively enumerating over the possible inputs would correspond to $n = \exp(O(T))$ experts, which would result in prohibitively large regret.

**Paper organization.** The remainder of this paper is structured as follows. Section 2 presents our main theoretical and empirical contributions. Section 3 introduces our spherical hierarchical region decomposition algorithm, establishes regret bounds, and develops the underlying mathematical framework. Section 4 demonstrates the practical effectiveness of our approach through experiments on real-world datasets. The appendix then provides comprehensive supporting material: Appendix A surveys related literature in online learning and low-rank approximation; Appendix B establishes necessary notation and background; Appendix C details the incremental coreset construction procedure that enables our algorithm's computational efficiency; Appendix D proves computational hardness results connecting online and offline approximation; Appendix E extends our core algorithmic framework to the weighted low-rank approximation setting; Appendix F contains complete proofs of all theoretical results; and Appendix G presents additional experimental validation on both synthetic and real-world datasets.

## 2 OUR CONTRIBUTIONS

While our focus is on online low-rank approximation, our results naturally extend to the more general setting of *weighted low-rank approximation* (WLRA), where each entry of the data matrix $\mathbf{A} \in \mathbb{R}^{n \times d}$ is assigned a non-negative weight $W_{i,j}$, and the goal is to find a rank-$k$ factorization $\mathbf{UV}$ minimizing the weighted Frobenius norm

$$\|\mathbf{W} \circ (\mathbf{A} - \mathbf{UV})\|_F^2 := \sum_{i=1}^{n} \sum_{j=1}^{d} W_{i,j} \left(A_{i,j} - (\mathbf{UV})_{i,j}\right)^2,$$

where $\circ$ denotes the Hadamard product. This formulation captures scenarios in which some entries are more significant, noisy, or costly to approximate than others, including recommendation systems with varying confidence in ratings, robust matrix completion, and other applications with heterogeneous importance. Classical low-rank approximation is the special case with uniform weights $W_{i,j} = 1$, but in practice, outlier or high-variance columns can dominate the approximation. WLRA balances such effects via entry-specific weights, e.g., inversely proportional to variance, enabling low-rank factors to capture the true underlying structure and improving interpretability and predictive performance (Gillis & Glineur, 2011). Weighted low-rank approximation is APX-hard (Gillis & Glineur, 2011), highlighting its computational difficulty. For further background and a comprehensive overview of weighted low-rank approximation techniques and applications, we refer the reader to the survey by Srebro & Jaakkola (2003).

**Information-theoretic online guarantees via multiplicative weights.** We introduce a novel online algorithm grounded in the Multiplicative Weights Update Algorithm (MWUA), tailored to the

weighted low-rank approximation setting. The algorithm operates over a hierarchy of candidate rank-$k$ bases derived from a structured decomposition of the unit sphere $\mathbb{S}^d$, allowing it to maintain diverse options for approximating incoming data points. At each time step, the algorithm selects a basis according to a probability distribution weighted by the historical performance of each candidate, effectively balancing exploration and exploitation in high-dimensional subspace selection. This design enables us to establish *information-theoretic regret bounds* of $\tilde{O}(\sqrt{T})$, where the notation suppresses poly-logarithmic dependencies on $T$ and polynomial dependencies on $k$ and $d$. While the full algorithm is not computationally efficient, it serves as a rigorous benchmark, providing insight into the fundamental limits of online approximation in the APX-hard regime of weighted low-rank approximation. Beyond the formal bounds, this contribution underscores the potential of combining multiplicative weights with geometric decompositions to adaptively manage candidate subspaces in streaming environments.

**Hardness results and offline-to-online connections.** Building on this theoretical foundation, we explore the interplay between online algorithms and classical offline computational complexity. Specifically, we construct a formal reduction showing that any online algorithm with per-step runtime $f(t, k, d)$ can be transformed into an offline algorithm that achieves additive error $\varepsilon$ in time polynomial in $n$, $k$, $d$, $\frac{1}{\varepsilon}$, and $f(t, k, d)$. This reduction implies a striking consequence: a polynomial-time online algorithm with sublinear regret would yield a fully polynomial-time approximation scheme (FPTAS) for offline weighted low-rank approximation, which contradicts known APX-hardness results (Gillis & Glineur, 2011). This connection provides a precise computational rationale for why efficient, fully optimal online algorithms are unlikely to exist and motivates the design of algorithms that trade off between achievable regret guarantees and feasible computation. Moreover, it situates online low-rank approximation within a broader theoretical framework connecting online learning, approximation hardness, and high-dimensional optimization, highlighting the subtle constraints imposed by sequential data arrival and weighted objectives.

**Empirical evaluation and adaptive hierarchical strategies.** Complementing our theoretical results, we present an empirical study that demonstrates the practical utility of our approach. We implement a simplified hierarchical decomposition in which regions are split along randomly generated orthogonal hyperplanes, forming a coarse yet effective geometric structure over the unit sphere. Despite this simplification relative to the more sophisticated angular decomposition used in the theoretical analysis, our MWUA-based algorithm consistently outperforms standard baselines across a variety of synthetic and real-world datasets. The experiments reveal several key insights: the algorithm robustly adapts to the intrinsic structure of data distributions, maintains low cumulative projection loss, and efficiently updates the approximation in a streaming setting. These results highlight the importance of exploiting geometric information when designing online low-rank approximation algorithms and suggest that full hierarchical decompositions could provide further improvements in both accuracy and computational efficiency. More broadly, the experiments illustrate how adaptive online algorithms can enable real-time analytics for high-dimensional streaming applications such as recommendation systems, fraud detection, and adaptive learning pipelines, where recomputing full singular value decompositions is infeasible.

Together, our contributions provide a comprehensive perspective on online weighted low-rank approximation. We establish both the limits of what is theoretically achievable and practical strategies for approaching these limits, bridging information-theoretic guarantees, computational complexity, and empirical performance. Our work not only advances the fundamental understanding of online matrix approximation under sequential constraints but also provides actionable guidance for designing adaptive, scalable algorithms capable of handling large-scale, high-dimensional data streams.

## 3 ALGORITHM

In this section, we present our algorithm for online low-rank approximation with spherical hierarchical refinement. The method combines two key ideas: (i) an adaptive spherical partitioning of the unit ball that generates a hierarchical set of candidate centroids, and (ii) a multiplicative weights scheme that treats these centroids as experts for constructing rank-$k$ approximations. The latter is presented in Algorithm 1, while the former is given in Algorithm 2. At each round, the algorithm updates a lightweight coreset with the new data point, refines the spherical decomposition only where needed, and adjusts expert weights based on observed projection loss. The selected basis is drawn according

to these weights, ensuring that the algorithm balances exploration of new regions with exploitation of historically accurate ones. This design guarantees sublinear regret relative to the best fixed low-rank subspace in hindsight, while maintaining computational and memory efficiency through the adaptive coreset and hierarchical refinement.

Intuitively, the spherical HRD regret minimization algorithm maintains a collection of simple candidate subspaces, represented by centroids of spherical regions, and adaptively learns which ones best capture the incoming data. Each region of the hierarchical spherical decomposition acts as an "expert," producing a basis vector candidate. At each time step, the algorithm forms a probability distribution over these experts using multiplicative weights: experts that previously explained the data well (low projection loss) are given higher weight, while poor-performing ones are down-weighted. The chosen rank-$k$ basis is then drawn from this distribution, ensuring that the algorithm balances exploration of new regions with exploitation of reliable ones. The resulting regret guarantee means that, over time, the algorithm performs nearly as well as the best fixed low-rank subspace in hindsight. The coreset plays the role of a compressed memory, allowing the algorithm to retain only the most informative vectors when making updates, which keeps computation efficient.

The spherical HRD update step determines how the partition of the sphere evolves as data accumulate. Given a new point, the algorithm checks whether the region containing it is too large or heterogeneous to represent the point well. If so, that region is refined by splitting it into smaller regions along angular coordinates, much like recursively bisecting the sphere. This adaptive refinement ensures that only regions where the data distribution is complex are split, so the tree structure becomes more detailed in dense or high-variance areas. The effect is a coreset that is spatially adaptive: it provides fine resolution only where necessary, while leaving sparse regions coarsely represented. This hierarchical structure allows the regret minimization algorithm to focus computational effort where it matters most.

---

**Algorithm 1** Spherical HRD Regret Minimization

---

**Input:** $\varepsilon_{\mathrm{hrd}}$, $x_1, \ldots, x_T$, learning rate $\eta$, rank $k$
1: Initialize spherical HRD tree $H_0$ with root region
2: Initialize coreset $S^{(0)} \leftarrow \emptyset$ and MTMW weights $w_0(p) = M(p)$
3: **for** $t = 1$ to $T$ **do**
4:     Update coreset $S^{(t)}$ with incoming vector $x_t$
5:     Apply refinement criterion and update HRD tree $H_t$
6:     Compute expert set $\mathcal{E}_t$ from active leaf centroids
7:     Compute probabilities $p_t(e)$ proportional to weights $w_{t-1}(e)$
8:     Select rank-$k$ expert basis $C_t = \{c_1, \ldots, c_k\}$ according to probabilities $p_t$
9:     Receive $x_t$ and incur projection loss $\ell(C_t, x_t)$
10:    Compute loss $\ell(e, x_t)$ for each expert $e \in \mathcal{E}_t$
11:    Update weights $w_t(e)$ using multiplicative rule with losses

---

**Algorithm 2** Spherical HRD Update Step

---

**Input:** $t$, $R_{t-1}$, $x_t \in S^d$, $\varepsilon_{\mathrm{hrd}}$
1: Let $q(\cdot)$ be the refinement criteria for $x_t$ at $t$;
2: $R_t \leftarrow \emptyset$
3: $U_t \leftarrow R_{t-1}$
4: **while** $U_t \neq \emptyset$ **do**
5:     Pick and Remove a region $R$ from $U_t$
6:     **if** q(R) **then**
7:         $R_t \leftarrow R_t \cup \{R\}$
8:     **else**
9:         $H \leftarrow \mathrm{split}(R)$                  ▷Halve along all angular coordinates
10:       $U_t \leftarrow U_t \setminus R \cup H$
11: **return** $R_t$

---

We now present the main theorem and defer the proof to the appendix.

**Theorem 3.1.** *Algorithm 1 has a regret of* $O\left(k\log\left(\frac{kT^3\sqrt{d}}{\varepsilon^2}\right)\sqrt{dT}\right) + \varepsilon\mathcal{C}$ *and a runtime of* $T\cdot$
$\tilde{O}\left(\frac{\sqrt{d}k^2T^3\log T}{\varepsilon^2}\right)^{kd}$.

## 3.1 Spherical Hierarchal Region Decomposition

A spherical region decomposition is a partition $\mathcal{R} = \{R_1, \ldots, R_\tau\}$ of the surface of a $d-$ *dimensional* unit ball, $\mathbb{S}^d$, where each part $R_i$ is referred to as a region. A spherical hierarchal region decomposition is a sequence of region decompositions $\{\mathcal{R}_1, \ldots, \mathcal{R}_\tau\}$ where each $\mathcal{R}_t$ is a refinement of $\mathcal{R}_{t-1}$, so that for all $\tau \in [t]$ and regions $R \in \mathcal{R}_\tau$, there exists a region $R' \in \mathcal{R}_{\tau-1}$ such that $R \subseteq R'$. Since a spherical hierarchical region decomposition $\mathcal{H} = \{\mathcal{R}_1, \ldots, \mathcal{R}_t\}$ only partitions existing regions, then we can naturally define a tree structure $T_\mathcal{H}$. Specifically, there is a node $T_\mathcal{H}$ for each region of each decomposition $\mathcal{R}_\tau$. There is an edge from the node representing region $R$ to the node representing region $R'$ if $R \subseteq R'$ and there exists a $\tau$ such that $R \in \mathcal{R}_\tau$ and $R' \in \mathcal{R}_{\tau-1}$. We slightly abuse notation and use $R$ to refer to the node corresponding to a region $R$. The bottom-level spherical decomposition is the decomposition induced by the leaves of the tree.

Given a spherical hierarchical decomposition $\mathcal{H}_t = \{\mathcal{R}_1, \ldots, \mathcal{R}_t\}$ and a set of points $S \subseteq \mathbb{S}^d$ of size $k$, we define the *representative regions* of $S$ in $\mathcal{H}_t$ as a sequence of multisets $\{\widetilde{R}_\tau\}_{\tau=1}^t$, where $\widetilde{R}_\tau = \{R_{\in\mathcal{R}_\tau} \mid \text{there exists } s \in S : s \in R\}$ with multiplicities defined with respect to $S$. These representative regions correspond to a path in $T_\mathcal{H}$.

Let $\mathbb{S}^d$ denote the unit sphere equipped with the uniform surface probability measure $\sigma$. For a measurable region $R \subseteq \mathbb{S}^d$, define the mean vector

$$v(R) = \int_{x \in R} x\, d\sigma(x) = \mathbb{E}_{x \sim \text{Unif}(R)}[x] \in \mathbb{R}^d.$$

The spherical centroid of $R$ is then given by

$$\text{centroid}(R) = \frac{v(R)}{\|v(R)\|_2} \in \mathbb{S}^d,$$

where $\|\cdot\|_2$ denotes the Euclidean norm in $\mathbb{R}^d$. As our regions will always be non-empty, we can ignore the degenerate case where $v(R) = 0$.

We then define the *approximate centers* of $S$ induced by $\mathcal{H}_t$ as the sequence of multisets $\{\widetilde{S}_\tau\}_{\tau=1}^t$, where $\widetilde{S}_\tau = \{\text{centroid}(R) \mid R \in \widetilde{\mathcal{R}}_\tau\}$.

## 3.2 Adaptive Spherical Hierarchical Region Decomposition

Given a sequence of points in $\mathbb{R}^d$, we describe an algorithm which maintains a hierarchal decomposition of the surface $\mathbb{S}^d$ of a $d$-dimensional hypersphere. We first define $d$-dimensional polar coordinates as follows. A point $x \in \mathbb{S}^d \subset \mathbb{R}^d$ can be represented in $d-1$-dimensional polar coordinates $x = (\theta_1, \theta_2, \ldots, \theta_{d-1})$, where the coordinates are defined recursively as

$$x_1 = \cos(\theta_1), \qquad x_2 = \sin(\theta_1)\cos(\theta_2), \qquad , \ldots, \qquad x_d = \sin(\theta_1)\cdots\sin(\theta_{d-2})\sin(\theta_{d-1}),$$

with ranges

$$0 \leq \theta_i \leq \pi \quad \text{for } 1 \leq i \leq d-2, \qquad 0 \leq \theta_{d-1} < 2\pi.$$

Using this coordinate system, we will define each region $R \subseteq \mathbb{S}^d$ in the hierarchical decomposition by a product of intervals in the angular coordinates:

$$R = [\theta_1^{\min}, \theta_1^{\max}] \times [\theta_2^{\min}, \theta_2^{\max}] \times \cdots \times [\theta_{d-1}^{\min}, \theta_{d-1}^{\max}].$$

Observe that bisection along these angular intervals defines the refinement of the hierarchical decomposition at each level. The spherical diameter $\Delta_R$ can then be measured as the maximum great-circle distance between any two points in $R$, which can be approximated from the ranges of the angular coordinates. Formally, we have $\Delta_R = \sup_{x,y \in R} \text{dist}_{\mathbb{S}^d}(x, y)$, where $\text{dist}_{\mathbb{S}^d}(x, y)$ denotes the geodesic (great-circle) distance between $x$ and $y$ on the unit sphere, i.e., $\text{dist}_{\mathbb{S}^d}(x, y) = \arccos(\langle x, y \rangle), \quad x, y \in \mathbb{S}^d.$

Given a sequence of points on the unit hypersphere $\mathbb{S}^d \subset \mathbb{R}^d$, we describe an algorithm that maintains a hierarchical region decomposition with regions defined via $d$-dimensional polar coordinates ($r = 1$) as follows. Let $\varepsilon_{\mathrm{hrd}} > 0$ be a parameter such that $\frac{\varepsilon_{\mathrm{hrd}}}{2T^3 \cdot \sqrt{d}}$ is a power of 2. This requirement allows us to define an implicit angular grid with angular resolution $\delta_T = \frac{\varepsilon_{\mathrm{hrd}}}{2T^3}$, such that the full spherical grid can be constructed from a single region covering the entire sphere by repeated bisection in all angular coordinates. Denote the angular resolution at level $t$ by $\delta_t = \frac{\varepsilon_{\mathrm{hrd}}}{2T^3}$. We refer to this implicit angular grid, together with the tree structure induced by the successive bisections, as the *full spherical grid* and the *full spherical grid tree*.

Consider a time $t$, a region $R \in \mathcal{R}_t$, and a point $x \in \mathbb{S}^d$. Denote the spherical diameter of $R$ by $\Delta_R$, as defined by the supremum of great-circle distances between points in $R$, and let $r = \min_{p \in R} \mathrm{dist}_{S^d}(p, x)$ be the geodesic distance between $x$ and $R$, so that $r = 0$ if $x \in R$.

We define the *refinement criterion* induced by $x$ at time $t$ as $q(R)$, which is $\texttt{true}$ if and only if the spherical diameter of $R$ satisfies $\Delta_R \leq \max\left(\frac{\varepsilon_{\mathrm{hrd}} \cdot r}{2}, \delta_t\right)$. At a given time $t$, a new unit vector $x_t$ is received, and the hierarchical region decomposition obtained at the end of time $t - 1$, $\mathcal{H}_{t-1}$, is refined by successively bisecting any region $R$ for which $q(R) = \texttt{false}$ along one or more angular coordinates, until all resulting regions satisfy the refinement criteria induced by all vectors $x_1, \ldots, x_t$ at their corresponding insertion times. This procedure guarantees that the spherical hierarchical decomposition maintains all angular intervals small enough relative to both the local geodesic distance from inserted points and the target resolution $\delta_t$.

**Lemma 3.2.** *Consider the hierarchal region decomposition $\{\mathcal{R}_1, \ldots, \mathcal{R}_t\}$ produced by the algorithm at any time $t$. Consider a region $R \in \mathcal{R}_{t-1}$. Then either region $R$ belongs to $\mathcal{R}_t$, or each child region of $R$ in $\mathcal{R}_t$ has angular diameter at most $\frac{4\pi}{2^t}$.*

We next bound the number of times a single region can be refined along a chain of nested regions. Consider a sequence of regions $\{R_t \in \mathcal{R}_t\}_{t=1}^{T}$ such that $R_{t+1} \subseteq R_t$ for all $t$. Let $\Lambda$ denote the number of strict refinements in this sequence, i.e., $\Lambda = |\{t \mid R_{t+1} \neq R_t\}|$. Since the algorithm never refines a region once its angular diameter is below $\delta_t$, the total number of possible refinements along such a chain is logarithmically bounded in the ratio between the initial and terminal diameters. Combining this stopping rule with the upper bound on diameters from Lemma 3.2, we have:

**Corollary 3.3.** *For any sequence of nested regions of length $T$, we have*

$$\Lambda \leq -\log\left(\frac{\delta_t}{\sqrt{d}}\right) = -\log\left(\frac{\varepsilon_{\mathrm{hrd}}}{2T^3 \sqrt{d}}\right).$$

**Lemma 3.4.** *Let $x_1, \ldots, x_n \in \mathbb{S}^d$ be a stream of unit vectors. The number of* new regions *added at any step $t$ is at most $\left(\frac{2\pi\sqrt{d}}{\delta_T}\right)^{d-1} \log\left(\frac{2T^3\sqrt{d}}{\varepsilon_{\mathrm{hrd}}}\right)$. Hence, the total number of regions after $N$ steps is at most $N\left(\frac{2\pi\sqrt{d}}{\delta_T}\right)^{d-1} \log\left(\frac{2T^3\sqrt{d}}{\varepsilon_{\mathrm{hrd}}}\right)$.*

**Corollary 3.5.** *Let $R \in \mathcal{R}_t$ be any region at step $t$, and let $S = \{R' \in \mathcal{R}_{t+1} \mid R' \subseteq R\}$ be the set of regions that refine $R$ in the next time step. Then the maximum branching factor $\beta$ satisfies*

$$\beta := \max_{t,R} |S| \leq \left(\frac{2\pi\sqrt{d}}{\delta_T}\right)^{d-1} \log\left(\frac{2T^3\sqrt{d}}{\varepsilon_{\mathrm{hrd}}}\right),$$

*due to Lemma 3.4. Furthermore, for sufficiently large $T$, we have $\beta \leq d \cdot \Lambda$, where $\Lambda$ is the maximum number of refinements along any path in the hierarchical decomposition.*

Having bounded the branching factor of the hierarchical decomposition, we next show that this structural control also ensures stability of the loss across nearby bases within the same region.

**Lemma 3.6.** *Consider an instance of the online low-rank approximation problem where a new unit vector $x_t \in \mathbb{R}^n$ arrives at each time step $t$, and let the spherical hierarchical region decomposition with parameter $\varepsilon_{\mathrm{hrd}}$ produce the regions $\mathcal{R}_t$ at step $t$. Let $\varepsilon_{\mathrm{hrd}} > 0$, and consider two multisets of rank-$k$ basis vectors $S = \{u_1, \ldots, u_k\}, S' = \{u_1', \ldots, u_k'\}$, with $u_i, u_i' \in \mathbb{R}^n$ lying on the unit sphere for all $i \in [k]$.*

*Suppose that for each $i \in [k]$, the pair $(u_i, u_i')$ lies within the same spherical region of the decomposition at step $t$. Then, for all $1 \leq \tau < t$,*

$$\ell(S', x_\tau) \leq (1 + \varepsilon_{\mathrm{hrd}})\,\ell(S, x_\tau) + \frac{\varepsilon_{\mathrm{hrd}}}{\tau^5}.$$

*In other words, for two points in the same region, the loss of $S'$ on any previously seen point is at most $(1 + \varepsilon_{\mathrm{hrd}})$ times the loss of $S$, up to a small additive term that decreases over time as $\frac{\varepsilon_{\mathrm{hrd}}}{\tau^5}$.*

We now extend this lemma to the low-rank approximation setting. Given a collection of $k$ unit vectors $U = \{u_1, \ldots, u_k\}$ spanning the candidate rank-$k$ subspace, and a spherical hierarchical region decomposition $\mathcal{H} = \{\mathcal{R}_1, \ldots, \mathcal{R}_t\}$, we associate with $U$ a sequence $\{\widetilde{U}_1, \ldots, \widetilde{U}_t\}$ of approximate bases induced by $\mathcal{H}$. Specifically, for each $u_i \in U$, we define its approximation at step $t$ to be the centroid $v_i$ of the region in $\mathcal{R}_t$ containing $u_i$. Note that $\widetilde{U}_t = \{v_1, \ldots, v_k\}$ is a multiset of unit vectors, which spans an approximate subspace $V_t$.

The next lemma follows directly from applying Lemma 3.6 to each such approximate basis $\widetilde{U}_t$ in place of $U$, and summing the resulting error bounds over all steps $t$.

**Lemma 3.7.** *Let $S^* = \{u_1, \ldots, u_k\}$ be the optimal set of $k$ basis vectors in hindsight for the stream $x_1, \ldots, x_T \subseteq \mathbb{R}^n$, and let $\tilde{S}_t = \{\tilde{u}_{t,1}, \ldots, \tilde{u}_{t,k}\}$ denote the approximate basis vectors induced by the spherical hierarchical decomposition with parameter $\varepsilon_{\mathrm{hrd}}$, where each $\tilde{u}_{t,i}$ is the centroid of the region in $\mathcal{R}_t$ containing $u_i$. Then the cumulative reconstruction loss of the approximate bases satisfies*

$$\sum_{t=1}^{T} \ell(\tilde{S}_{t+1}, x_t) \leq (1 + \varepsilon_{\mathrm{hrd}}) \sum_{t=1}^{T} \ell(S^*, x_t) + 2\varepsilon_{\mathrm{hrd}},$$

*where $\mathcal{C} = \sum_{t=1}^{T} \ell(S^*, x_t)$ is the optimal offline reconstruction cost.*

As Corollary 3.3 shows, each basis vector $u_i$ can move to a new region centroid at most $\Lambda$ times, since each of the $k$ regions containing $u_i$ can be refined at most $\Lambda$ times. Hence, the approximate basis $\tilde{S}_{t+1}$ differs from $\tilde{S}_t$ at most $k \cdot \Lambda$ times over the entire stream. Moreover, since the instantaneous loss $\ell(S, x_t)$ is bounded by $d$, combining this with Lemma 3.7 yields the following corollary.

**Corollary 3.8.** *For the optimal set of basis vectors in hindsight $S^*$ and the approximate basis vectors $\tilde{S}_t$ induced by the Spherical Hierarchical Region Decomposition at time step $t$, for an unweighted stream $x_1, \ldots, x_T$ we have*

$$\sum_{t'=1}^{T} \ell(\tilde{S}_{t'}, x_{t'}) \leq (1 + \varepsilon_{\mathrm{hrd}})\mathcal{C} \;+\; k \cdot \Lambda + 2\varepsilon_{\mathrm{hrd}},$$

## 3.3 MTMW - MWUA FOR TREE STRUCTURED EXPERTS

We present an algorithm, which we call *Mass Tree MWUA (MTMW)*, that achieves low regret in the setting of *Prediction from Expert Advice*, adapted to the online low-rank approximation problem. Our presentation follows the hierarchical decomposition framework of Cohen-Addad et al. (2021) for online clustering, combined with the regret analysis of Arora et al. (2012), and we adapt the classical Multiplicative Weights Update Algorithm (MWUA) to this setting to obtain an analogous regret bound. The set of experts is structured according to the spherical hierarchical region decomposition (HRD) in polar coordinates, as previously described. Let $\ell$ denote a bounded loss function on the unit sphere, with $\ell(x, y) \in [-1, 1]$ for all $x, y \in \mathbb{S}^d$. Consider a spherical HRD $\mathcal{H}_T$ of depth $T$, where each leaf corresponds to a region of minimum angular diameter $\delta_T$. Each vertex of the decomposition corresponds to a region, and the set of experts is defined as all sequences of nested regions from the root to a leaf:

$$\mathcal{P}(\mathcal{H}_T) = \{(R_1, \ldots, R_T) : R_1 \text{ is the root region}, \; R_T \text{ is a leaf region}\}.$$

For a path $p = (R_1, \ldots, R_T) \in \mathcal{P}(\mathcal{H}_T)$, the prediction at step $t$ is taken to be the centroid of region $R_t$, and the cumulative loss of $p$ on a sequence of arriving unit vectors $X_{1:T} = \{x_1, \ldots, x_T\}$ is

$$\ell(p, X_{1:T}) = \sum_{t=1}^{T} \ell(\mathrm{centroid}(R_t), x_t).$$

To initialize the weights, we associate a *mass* to each region in the decomposition. The root is assigned mass 1, and any other region $R$ with parent region $R'$ is assigned $M(R) = \frac{M(R')}{\deg(R')}$, where $\deg(R')$ is the number of children of $R'$. The mass of a path is defined as the mass of its terminal leaf region.

This mass structure allows the MTMW algorithm to perform multiplicative weight updates over the hierarchical expert set efficiently, exploiting the nested structure of the spherical HRD. Before proving the regret bound, we present a key lemma on the properties of the mass distribution and the relationship between region refinement and the maximum branching factor of the decomposition.

**Lemma 3.9.** *Let $v$ be a region in the spherical hierarchical decomposition, $\tilde{\mathcal{T}}$ a subtree rooted at $v$, and $\tilde{V}$ the set of leaf regions of $\tilde{\mathcal{T}}$. Then the mass of $v$ equals the sum of the masses of its leaf regions, i.e., $M(v) = \sum_{v' \in \tilde{V}} M(v')$.*

We now upper bound the expected regret of our algorithm.

**Theorem 3.10.** *Let $\mathcal{T}_t$ be the tree corresponding to the spherical hierarchical region decomposition at step $t$. Consider running the Mass Tree Multiplicative Weights Update Algorithm (MTMW) over the set of experts given by all root-to-leaf paths of the final tree, $\mathcal{P}(\mathcal{T}_t)$. Even if the tree is revealed adaptively up to depth $t$ at each time step, running MTMW is possible provided each path $p \in \mathcal{P}(\mathcal{T}_t)$ is initialized with weight $M(p)$. Then, the regret of MTMW with respect to any path $p$ is bounded by*

$$Regret \leq \sqrt{-T \ln M(p)}.$$

*Moreover, the algorithm can be implemented with time complexity $O(|\mathcal{T}_t|)$, i.e., proportional to the number of vertices in the tree.*

**Corollary 3.11.** *For the low-rank matrix approximation problem with loss bounded by $d$, the Mass Tree MWUA (MTMW) algorithm achieves a regret bound of $Regret \leq d\sqrt{-T \ln(M(p))}$, where $M(p)$ is the initial mass of the path corresponding to expert $p$, and the bound follows from using a normalized loss function.*

## 3.4 REGRET BOUND

Based on the above components of the algorithm, the regret bound for the full algorithm:

Before stating the main lemma in Lemma 3.12, we clarify the interpretation of the terms that appear in the bound. The multiplicative term captures the approximation errors introduced by the algorithm: $\varepsilon_c$ corresponds to the error incurred approximating basis vectors with the centroids of their spherical regions (canonical representations), while $\varepsilon_{\mathrm{hrd}}$ accounts for the error due to the hierarchical region decomposition. The sum of these two errors is scaled by $k$, the rank of the approximation, and $\Lambda$, the maximum refinement depth of the spherical HRD; the factor of $8$ arises from the geometric argument in the proof of Corollary 3.5. The additive term reflects the cost associated with maintaining $k$ experts across $\Lambda$ levels of refinement in the Mass Tree MWUA algorithm. Specifically, $d$ is the data dimensionality, and $k\Lambda$ bounds the cumulative regret incurred, as detailed in Corollary 3.3.

**Lemma 3.12.** *Let $\mathcal{H}$ be a spherical HRD with parameter $\varepsilon_{\mathrm{hrd}}$ that is constructed from $\{\mathcal{Q}_t\}_{t=1}^T$. Let $S^*$ be the best low-rank basis in hindsight, and let $\tilde{S}_t$ be the approximate basis induced by $\mathcal{H}$ at time $t$. Then, we have:*

$$\sum_{t=1}^T \ell(\tilde{S}_t, x_t) \leq (1 + \varepsilon_c + 8(\varepsilon_{\mathrm{hrd}} + \varepsilon_c)k\Lambda)\mathcal{C} + k\Lambda$$

*where $C = \sum_{t=1}^T \ell(S^*, x_t)$ is the optimal offline reconstruction cost.*

Consider the Spherical Hierarchical Region Decomposition $\mathcal{H}_t$ at step $t$, and its associated region tree $\mathcal{T}_{\mathcal{H}_t}$ as described in Section 3.1. We construct a *$k$-region tree* by taking a level-wise $k$-fold tensor product of $\mathcal{T}_{\mathcal{H}_t}$: each vertex at depth $t$ corresponds to a $k$-tuple of regions $(v_1, \ldots, v_k)$ at level $t$ of $\mathcal{T}_{\mathcal{H}_t}$, and a directed edge from $(v_1, \ldots, v_k)$ at level $t$ to $(u_1, \ldots, u_k)$ at level $t+1$ exists if and only if $(v_i, u_i)$ is an edge in $\mathcal{T}_{\mathcal{H}_t}$ for every $i \in [k]$. We then define the corresponding *$k$-basis tree* (or $k$-tree), whose vertices correspond to the centroids of the regions rather than the regions themselves. Concretely, a vertex $(v_1, \ldots, v_k)$ in the $k$-region tree corresponds to a vertex

$[\mu(v_1), \ldots, \mu(v_k)]$ in the $k$-tree, where $\mu(v_i)$ denotes the centroid of region $v_i$. Each set of $k$ basis vectors $S = \{u_1, \ldots, u_k\}$ can be associated with a path in the $k$-region tree representing the regions containing each $u_i$, and the corresponding approximate basis vectors $\tilde{S}_t$ are associated with the path in the $k$-tree through the centroids of these regions.

In this framework, the Mass Tree MWUA (MTMW) algorithm assigns a mass $M(v)$ to each vertex $v$ in the $k$-tree, recursively defined as in Lemma 3.9. The mass of a path $p$ through the $k$-tree reflects the combined probability weight of choosing the sequence of approximate basis vectors along that path. Lemma 3.12 guarantees that there exists a path $p^*$ in the $k$-tree corresponding to the optimal set of basis vectors $S^*$ in hindsight, such that the cumulative loss of the sequence of centroids along $p^*$ is close to the offline optimal reconstruction cost $\mathcal{C}$.

**Lemma 3.13.** *Let $p^*$ denote the path in the $k$-tree corresponding to the best set of basis vectors in hindsight. $\Lambda$ is the maximum refinement depth of the HRD and $\beta$ bounds the logarithm of the maximum branching factor of any region in the decomposition, as defined as in Corollary 3.3 and Corollary 3.5, respectively. Then we have $-\ln M(p^*) \leq k^2 \Lambda \beta$.*

Here, $-\ln M(p^*)$ quantifies the "effective number of experts" in the MWUA analysis, i.e., the cumulative weight spread across all possible sequences of approximate basis vectors in the $k$-tree. This upper bound ensures that MTMW achieves low regret relative to the optimal set of basis vectors $S^*$ in hindsight. This concludes all the components to prove our main result of Theorem 3.1.

# 4 EXPERIMENTS ON MNIST DATASET

We use the MNIST (Modified National Institute of Standards and Technology) 784 dataset (LeCun et al., 1998), which has 70,000 observations and 784 features. MNIST is a dataset widely used in machine learning, where each example is a handwritten digit and the features are the pixel intensity values of the digit image. This dataset was accessed through Kaggle.

**Experimental setup.** We perform dimensionality reduction on the MNIST dataset, reducing to $d = 50$ dimensions with TruncatedSVD. This gives us a low rank approximation of the original matrix with the top 50 singular values. The reduced vectors are then also normalized to unit length. The HRD algorithm tests on target basis vector rank $k = 10$, $k = 15$, and $k = 20$, splitting threshold $d_{split} = 15$, minimum leaf size $n_{min=20}$, maximum leaf size $n_{max} = 100$, learning rate $\eta = 0.5$, and refinement parameter $\varepsilon_{\mathrm{hrd}} = 0.1$. We stream 500 data points and compare performance to the same baseline non-adaptive net introduced in the synthetic experiments.

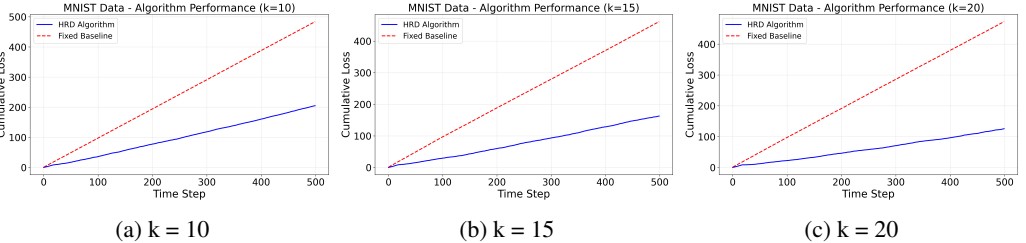

(a) k = 10        (b) k = 15        (c) k = 20

Fig. 1: Cumulative Loss of HRD Algorithm and Fixed Baseline Net over 500 Data Points for MNIST Dataset with different values of k

**Results and discussion.** Our results in Figure 1 show that the HRD algorithm outperforms the fixed baseline by a wide margin in all three cases, with a percent decrease in reconstruction loss of $57.64\%$ in the rank-10 case, $64.75\%$ in the rank-15 case, and $73.42\%$ in the rank-20 case. Because the MNIST dataset presents naturally occurring patterns from handwritten digits, these results indicate that the HRD algorithm successfully adapts to patterns presented in the dataset, demonstrating applicability beyond controlled, synthetic situations even in higher dimensions. Moreover, as expected, the improvements delivered by the HRD algorithm increased as we increased the dimension of the basis vectors. Notably, this experiment ran for significantly longer than the synthetic experiments despite having only 500 time steps compared to 1000. This increased computational cost reflects the algorithm's adaptive behavior when encountering complex, real-world data structures that require more sophisticated partitioning decisions.

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

## A    RELATED WORK

**Low-rank approximation.** Low-rank approximation has a rich history in both linear algebra and machine learning, serving as a foundational technique for data compression, dimensionality reduction, and uncovering latent structure. The seminal result of Eckart & Young (1936) established that the optimal rank-$k$ approximation of a matrix in the Frobenius norm is obtained via truncated singular value decomposition (SVD). Building on this, a wide range of classical algorithms have been developed for efficiently computing low-rank approximations of static datasets. Randomized SVD

methods (Halko et al., 2011) exploit random projections to accelerate computation while preserving accuracy, and principal component analysis (PCA) (Jolliffe, 2002) remains a widely used method for identifying directions of maximal variance in high-dimensional data. These techniques are central to many modern applications, including collaborative filtering, natural language processing, and computer vision, where they reduce storage requirements, accelerate downstream computations, and improve interpretability.

The online or streaming setting, in which data arrives sequentially, introduces additional challenges. Unlike static datasets, where the full matrix is known prior, online low-rank approximation must maintain an accurate representation of the matrix as new rows or columns arrive, without recomputing the full SVD at each step. Incremental SVD updates (Brand, 2006) address this by algebraically modifying existing decompositions to account for additions or deletions of rows and columns, enabling efficient updates while retaining the low-rank structure. Similarly, online subspace identification from incomplete observations (Balzano et al., 2010) generalizes these ideas to sparse streaming matrices, which are common in recommendation systems and user-item interactions. In these settings, the matrix is only partially observed at each time step, and the algorithm must adaptively update the underlying low-dimensional subspace to provide accurate predictions.

Online PCA extends these ideas further, enabling incremental dimensionality reduction in applications with continuously arriving high-dimensional data. For instance, Ross et al. (2008) apply online PCA to visual tracking, processing frames sequentially, projecting each onto the current subspace, and updating mean, covariance, and principal components. Subsequent work (Feng et al., 2013) addresses the limitations of classical online PCA methods by incorporating robustness against outliers and corrupted observations, which frequently arise in real-world streaming datasets. These approaches demonstrate the importance of maintaining adaptability and resilience in online low-rank models, particularly when the data distribution is non-stationary or contains noise.

Collectively, these contributions highlight both the theoretical foundations and practical algorithms for online low-rank approximation. They motivate the need for methods that balance computational efficiency with adaptive accuracy, especially in high-dimensional, streaming, and noisy environments. Our work builds on these insights by integrating hierarchical geometric decomposition with multiplicative weights strategies, providing a framework that achieves provable regret guarantees while maintaining adaptability to evolving data distributions.

**Online learning with experts.** Online learning with experts and related sequential decision-making problems have been extensively studied in the machine learning and theoretical computer science communities (Cesa-Bianchi & Lugosi, 2006). In this framework, an algorithm repeatedly selects actions or "experts" from a predefined set, receives feedback in the form of losses, and aims to minimize its cumulative loss relative to the best fixed expert in hindsight. Early work assumed the existence of a "perfect expert" whose predictions were always correct. In this idealized scenario, a folklore "halving" algorithm achieves at most $\log_2 n$ mistakes by retaining the set of experts that have not yet erred and predicting based on majority vote.

Recognizing that a perfect expert rarely exists in practice, Littlestone & Warmuth (1994) introduced the randomized weighted majority (RWM) algorithm. Here, predictions are sampled according to a probability distribution proportional to the experts' weights, which are updated multiplicatively after each round. This yields $O(\sqrt{T \log n})$ regret and is asymptotically optimal (Cover, 1966). More generally, the Multiplicative Weights Update (MWU) framework extends these ideas to arbitrary loss functions and has found applications in boosting (e.g., AdaBoost (Freund & Schapire, 1997)), approximately solving zero-sum games (Freund & Schapire, 1999), and efficiently approximating linear and semi-definite programs (Clarkson, 1995; Plotkin et al., 1995; Garg & Könemann, 2007). Variants such as Follow the Perturbed Leader (FTPL) and Follow the Regularized Leader (FTRL) introduce random perturbations or regularization to improve stability and computational efficiency in structured problems (Kalai & Vempala, 2005; McMahan, 2011). For an in-depth survey of MWUA and its widespread applications in online optimization, game theory, and combinatorial learning, see (Arora et al., 2012).

A particularly relevant extension is the work on online learning with low-rank experts by Hazan et al. (2016), which exploits hidden low-rank structure in the expert loss matrix. In this setting, although the nominal number of experts may be very large, the effective dimensionality of the loss matrix is small, allowing regret bounds to scale with the rank rather than the total number of experts. This

insight is closely aligned with our online low-rank approximation problem: in our formulation, each candidate low-rank basis can be interpreted as an "expert", and the losses correspond to squared residuals of projections. However, a key distinction in our work is that the expert set itself evolves over time through incremental coreset updates and hierarchical decomposition of the space, rather than being fixed in advance. This adaptive structure introduces additional challenges in maintaining low regret while efficiently selecting high-quality bases.

Variants of MWUA have also been developed to handle hierarchical or tree-structured expert sets. Notably, the Mass Tree MWUA (MTMW) framework by Cohen-Addad et al. (2021) introduces a hierarchical organization of experts, allowing multiplicative updates to propagate efficiently along the tree structure. This hierarchical approach provides both computational advantages and more refined control over exploration versus exploitation across different scales of the expert set. Our algorithm builds directly on these ideas: we treat candidate low-rank bases as hierarchical experts organized according to a decomposition of the unit sphere, and we apply multiplicative weight updates at multiple levels of this hierarchy. By combining these hierarchical MWUA techniques with adaptive coreset selection, we can achieve theoretically meaningful regret guarantees in the challenging setting of online low-rank approximation, while also enabling practical performance improvements on high-dimensional streaming data.

Overall, these connections demonstrate that our approach not only leverages foundational insights from the classical MWUA and low-rank experts literature, but also extends them to a dynamic, geometrically structured expert space, bridging online learning theory with modern challenges in matrix approximation and streaming data analysis.

## B  PRELIMINARIES

**Notation and basic definitions.** We first restate some basic notations and definitions necessary for our results. We use the notation $[n]$ to represent the set $\{1, ..., n\}$ for some integer $n$. We typically use bold-font to denote matrices, whereas we use default-font variables to represent vectors and scalars. For a matrix $\mathbf{A} \in \mathbb{R}^{k \times d}$, we denote its squared Frobenius norm by

$$\|\mathbf{A}\|_F^2 = \sum_{i=1}^{k} \sum_{j=1}^{d} A_{ij}^2,$$

which is the sum of square of all entries of $\mathbf{A}$.

For a vector $x \in \mathbb{R}^d$, we denote the squared Euclidean norm by $\|x\|_2^2 = \sum_{j=1}^{d} x_j^2$. We use $\circ$ to denote the Hadamard (element-wise) product of matrices. For points $x, y \in S^d$, we define the geodesic (great-circle) distance as

$$\text{dist}_{S^d}(x, y) = \arccos(\langle x, y \rangle).$$

**Multiplicative weighs update algorithm.** For completeness, we briefly recall the standard Multiplicative Weights Update algorithm. For a learning rate $\eta \approx \frac{1}{\sqrt{T}}$, MWU guarantees low regret over $T$ rounds. Specifically:

**Theorem B.1.** *Consider $n$ experts with losses at most $\rho$ on each of $T$ rounds. The MWU algorithm, c.f., Algorithm 3, achieves expected regret $O\left(\rho\sqrt{T \log n}\right)$.*

---

**Algorithm 3** Multiplicative Weights Update (MWU)

---

**Input:** Learning rate $\eta$, expert losses $\{\ell_i(t)\}$ for $i \in [n]$, $t \in [T]$
**Output:** Sequence of expert selections
 1: Initialize cumulative losses: $w_i \leftarrow 0$ for all $i$
 2: **for** $t = 1$ to $T$ **do**
 3:    **for** $i = 1$ to $n$ **do**
 4:       update cumulative loss: $w_i \leftarrow w_i + \ell_i(t)$
 5:    Select expert $i$ with probability proportional to $\exp(-\eta w_i)$

---

**Problem definition.** We define the online low-rank approximation problem as follows. At each time step $t$, a row vector $x_t \in [0,1]^d \subseteq \mathbb{R}^d$ arrives, and we denote $\mathbf{X}_{1:t-1}$ the matrix formed by stacking the rows $\{x_1, \ldots, x_{t-1}\}$. The learning algorithm must output a rank-$k$ approximation $\mathbf{V}_t$ using only $\mathbf{X}_{1:t-1}$. We refer to $\mathbf{V}_t$ as the algorithm's hypothesis subspace at time $t$. For convenience, we define the *multiset representation* of this basis as

$$S_t := \{v_{t,1}, v_{t,2}, \ldots, v_{t,k}\},$$

where $S_t$ contains all column vectors of $V_t$. Viewing $\mathbf{V}_t$ as a multiset $S_t$ allows us to reason about the basis in terms of its constituent vectors, and to compare two bases $\mathbf{V}_t, \mathbf{V}_t'$ by comparing the corresponding multisets $S_t, S_t'$. With this representation in place, we can now define the instantaneous loss incurred by the algorithm at time t as

$$\ell(S_t, x_t) = \|x_t - \mathbf{V}_t(\mathbf{V}_t)^\top x_t\|_2^2$$

that is, the squared reconstruction error of $x_t$ under the approximation $\mathbf{V}_t$.

The total loss of the algorithm up to time $t$ is $L_t = \sum_{\tau=1}^{t} \ell(S_\tau, x_\tau)$. Let $S_t^*$ denote the multiset of the best rank-$k$ approximation to the data $\{x_\tau\}_{\tau=1}^{t-1}$. In hindsight, the optimal rank-$k$ solution after $T$ steps is denoted $S_{T+1}^*$. The loss of the best rank-$k$ solution in hindsight after $T$ steps is denoted as $\mathcal{C}$ and is defined as $\mathcal{C} = \sum_{\tau=1}^{T} \ell(S_{T+1}^*, x_\tau)$. The regret is then defined as $\mathrm{Regret}(T) = L_T - \mathcal{C}$.

## C    CORESET CONSTRUCTION

The first part of our algorithm is to maintain the sets of basis vectors $\{S^{(t)}\}_{t \in [T]}$, which induces a $(1+\varepsilon)$-approximation to the best rank-$k$ approximation problem at each time $t \in [T]$. Moreover, each $S^{(t)}$ contains at most $O(k \cdot \log^3 T)$ basis vectors and the sequence $S^{(1)}, S^{(2)}, \ldots, S^{(T)}$ forms an *incremental coreset* for low-rank approximation. Specifically, it is monotone in the sense that $S_i^{(t-1)} \subseteq S_i^{(t)}$ for all $t$, and any data vector $x$ that is represented using a basis in $S_i^{(t)}$ at time $t$ remains represented by that same basis in all subsequent sets $S_i^{(\tau)}$ for $t \leq \tau \leq T$. To construct and maintain these incremental low-rank coresets, we use the algorithm of Braverman et al. (2020), whose performance guarantees are summarized in the following proposition, which follows directly from their work.

**Theorem C.1.** *(Braverman et al., 2020) There exists a randomized algorithm that constructs an incremental coreset for low-rank approximation with probability at least $1 - \frac{1}{T^5}$ while storing $O\left(\frac{k}{\varepsilon^2} \log^3 T\right)$ rows at any time and using nearly input-sparsity time.*

We remark that as stated in Braverman et al. (2020), the *online condition number*, defined as the ratio of the largest nonzero singular value to the smallest nonzero singular value of any intermediate matrix defined by the data stream must be bounded by some polynomial in $T$, in order to achieve $(1+\varepsilon)$-multiplicative approximation in the above stated guarantee. Such an online condition number bound may not necessarily hold even if each row of the input matrix is a vector with entries represented by $O(\log T)$ bits, normalized to a unit vector. In particular, there exist examples of *anti-Hadamard matrices* with dimension $n \times d$ but optimal low-rank cost as small as $\exp(-O(k))$. Hence, the online condition number can be as large as $\exp(O(k)) \cdot \mathrm{poly}(T)$, which would potentially result in a coreset with size $k^3 \cdot \mathrm{polylog}(T)$, since the number of rows sampled, as formally stated by Braverman et al. (2020), is $O\left(\frac{k}{\varepsilon^2} \log n \log^2 \kappa\right)$, where $\kappa$ is the online condition number. However, if we permit additive error by our coreset, then we can achieve the stated number of rows $O\left(\frac{k}{\varepsilon^2} \log^3 T\right)$ with an additional additive error $O(1)$, which suffices for our purposes.

## D    LOWER BOUND

Given the disappointing runtime of the grid-MWUA algorithm, one may wonder whether there is a way to avoid explicitly storing a weight for each of the exponentially many experts and speed up the MWUA algorithm. The following result gives evidence that it is unlikely that a significant speed-up is possible under complexity-theoretic assumptions. First, recall that weighted low-rank approximation is APX-hard (Gillis & Glineur, 2011), and the best known algorithms run in time exponential in at least one of the parameters $k$ or $d$.

**Theorem D.1.** *(Gillis & Glineur, 2011) Weighted low-rank approximation is NP-hard to approximate to within a multiplicative factor of $\left(1 + \frac{1}{\text{poly}(n)}\right)$.*

On the other hand, a consequence of the following statement is that for instances of weighted low-rank approximation with entries bounded in magnitude and optimal cost $C \geq \frac{1}{\text{poly}(n)}$, a per-round polynomial-time online algorithm would imply the existence of a fully polynomial-time approximation scheme.

**Theorem D.2.** *Suppose there exists an online weighted low-rank approximation algorithm $\mathcal{A}$ that achieves regret $\tilde{O}(T^{1-\alpha})$ and, at time $t$, runs in time $f(t, k, d)$. Then, for any $\varepsilon > 0$, there exists a randomized offline algorithm that, given an instance of weighted low-rank approximation, outputs a solution with cost at most $C + \varepsilon$ with constant probability and runs in time polynomial in $n, k, d, \frac{1}{\varepsilon}$, and $f(n, k, d)$.*

## E  EXTENSION TO WEIGHTED LOW-RANK APPROXIMATION

We now extend the framework to the weighted low-rank approximation (WLRA) setting. Let $\mathbf{W}$ denote the weight matrix and define $W = \max_{i,j} \mathbf{W}_{i,j}$. In this case, the projection loss for a vector $x_t$ with respect to the multiset representation $S_t$ of the rank-$k$ basis $\mathbf{V}_t$ is bounded by

$$\ell(S_t, x_t) \ \leq \ W^2 \left\| x_t - \mathbf{V}_t \mathbf{V}_t^\top x_t \right\|_2^2,$$

so every loss expression inherits an additional multiplicative factor of $W^2$.

This scaling affects the spherical hierarchical region decomposition (HRD), which must be refined more aggressively to accommodate high-weight points. Specifically, we adjust both the angular resolution and refinement criterion to

$$\delta_t = \frac{\varepsilon_{\text{hrd}}}{2T^3 W^2}, \qquad \Delta_R \leq \max\left(\frac{\varepsilon_{\text{hrd}} \cdot r}{2W^2}, \delta_t\right),$$

where $r$ is the geodesic distance from $x_t$ to region $R$. These modifications propagate through Lemmas 3.2, 3.4 and 3.6 and corollaries 3.3 and 3.5, introducing at most an additional factor of $W^2$. In effect, the HRD is refined proportionally to the weights while maintaining the same guarantees on regret and coreset size.

By contrast, the MTMW definitions themselves, including expert weights, update rules, and aggregation, are unaffected: the MWUA rule applies exactly as before. The only indirect impact is that finer HRD refinements increase the number of experts. This enlargement of the expert pool does not alter MWUA's mechanics but does appear in the regret bound through the $\ln M(p)$ term.

In summary, introducing weights leaves the MWUA dynamics unchanged while inducing finer HRD refinements. Consequently, the regret bound is scaled by a factor of $W^2$ and may include a larger poly-logarithmic term due to the increased number of experts. The runtime is likewise affected only through this growth in the expert pool, so the asymptotic structure of the algorithm remains the same; both regret and runtime scale smoothly with the weight magnitude $W$.

## F  MISSING PROOFS

### F.1  ADAPTIVE SPHERICAL HIERARCHICAL DECOMPOSITION

**Lemma 3.2.** *Consider the hierarchal region decomposition $\{\mathcal{R}_1, \ldots, \mathcal{R}_t\}$ produced by the algorithm at any time $t$. Consider a region $R \in \mathcal{R}_{t-1}$. Then either region $R$ belongs to $\mathcal{R}_t$, or each child region of $R$ in $\mathcal{R}_t$ has angular diameter at most $\frac{4\pi}{2^t}$.*

*Proof.* We instead show the decay of the spherical diameter under full bisection as follows. Let the root region (level 0) be represented by angular intervals

$$R^{(0)} = \prod_{i=1}^{d}[\theta_i^{0,\min}, \theta_i^{0,\max}],$$

and write $L_i^{(0)} = \theta_i^{0,\max} - \theta_i^{0,\min}$. Suppose the hierarchical decomposition is obtained by bisecting every angular interval at each refinement step. By the construction assumption, we bisect every angular interval at each level. Therefore after $t$ bisections each original interval length $L_i^{(0)}$ has been divided into $2^t$ equal sub-intervals, so the $i$-th angular interval length satisfies

$$L_i^{(t)} = \frac{L_i^{(0)}}{2^t},$$

and the maximum interval length $\delta_t = \max_i L_i^{(t)}$ satisfies $\delta_{t+1} = \frac{\delta_t}{2}$ and by induction, $\delta_t \leq \frac{\delta_0}{2^t}$.

Now, recall the chord-angle relation on the unit sphere. Specifically, for two points $x, y \in \mathbb{S}^d$ whose geodesic separation (angle) is $\alpha = \mathrm{dist}_{\mathbb{S}^d}(x, y)$, the Euclidean chord length satisfies

$$\alpha \leq \frac{\pi}{2} \cdot \|x - y\|_2.$$

Specifically, we have for all $\theta \in \left[0, \frac{\pi}{2}\right]$ that $\sin\theta \geq \frac{2}{\pi} \cdot \theta$ and thus

$$\|x - y\|_2 = 2\sin(\alpha/2) \geq \frac{2}{\pi} \cdot \alpha.$$

Now, for any two vectors in $\mathbb{R}^{d+1}$ the Euclidean distance is at most the $L_2$-norm of the vector of per-coordinate angular differences. In particular, by combining coordinate-wise differences and the fact each coordinate change is at most $L_i^{(t)}$, we obtain

$$\|x - y\|_2 \leq \sqrt{\sum_{i=1}^d (L_i^{(t)})^2} \leq \sqrt{d} \cdot \delta_t.$$

Fix a child $R'$ of $R$. Since $\Delta_{R'}$ is the supremum of the geodesic distances, then we have $\Delta_{R'} \leq$ geodesic distance is comparable to the chord length in the small-angle regime, we therefore have the angle-to-chord bound

$$\Delta_{R'} \leq \frac{\pi}{2} \cdot \sqrt{d}\,\delta_t \leq 2\delta_t.$$

Since $\delta_t \leq \frac{\delta_0}{2^t}$ and $\delta_0 = 2\pi$, then we have $\Delta_{R'} \leq \frac{4\pi}{2^t}$, as desired. $\qquad\square$

**Lemma 3.4.** *Let $x_1, \ldots, x_n \in \mathbb{S}^d$ be a stream of unit vectors. The number of* new regions *added at any step $t$ is at most $\left(\frac{2\pi\sqrt{d}}{\delta_T}\right)^{d-1} \log\left(\frac{2T^3\sqrt{d}}{\varepsilon_{\mathrm{hrd}}}\right)$. Hence, the total number of regions after $N$ steps is at most $N\left(\frac{2\pi\sqrt{d}}{\delta_T}\right)^{d-1} \log\left(\frac{2T^3\sqrt{d}}{\varepsilon_{\mathrm{hrd}}}\right)$.*

*Proof.* Consider a point $x_t \in \mathbb{S}^d$ arriving at step $t$. Let $R \in \mathcal{R}_{t-1}$ be a region containing $x_t$. Each region is rectangular in polar coordinates, with angular width $\Delta_R^{(k)}$ along coordinate $\theta_k$ for each $k \in [d-1]$. To satisfy the refinement criterion, $R$ is subdivided along each coordinate so that the maximum angular width does not exceed $\delta_T$.

Then the number of subregions along coordinate $\theta_k$ is at most $\lceil \frac{\Delta_R^{(k)}}{\delta_T} \rceil \leq \frac{2\pi}{\delta_T}$, since $\Delta_R^{(k)} \leq 2\pi$. Multiplying over all $d - 1$ coordinates gives

$$\gamma := \prod_{k=1}^{d-1} \left\lceil \frac{\Delta_R^{(k)}}{\delta_T} \right\rceil \leq \left(\frac{2\pi}{\delta_T}\right)^{d-1}.$$

By Corollary 3.3, the hierarchical decomposition has depth at most $\log\left(\frac{2T^3\sqrt{d}}{\varepsilon_{\mathrm{hrd}}}\right)$, so the total number of new regions added to accommodate $x_t$ is at most

$$\gamma \cdot \log\left(\frac{2T^3\sqrt{d}}{\varepsilon_{\mathrm{hrd}}}\right) \leq \left(\frac{2\pi\sqrt{d}}{\delta_T}\right)^{d-1} \log\left(\frac{2T^3\sqrt{d}}{\varepsilon_{\mathrm{hrd}}}\right),$$

accounting for the $\sqrt{d}$ factor converting coordinate widths to geodesic diameter. Summing over all $N$ points gives the claimed bound. $\qquad\square$

**Corollary 3.5.** *Let $R \in \mathcal{R}_t$ be any region at step $t$, and let $S = \{R' \in \mathcal{R}_{t+1} \mid R' \subseteq R\}$ be the set of regions that refine $R$ in the next time step. Then the maximum branching factor $\beta$ satisfies*

$$\beta := \max_{t,R} |S| \leq \left( \frac{2\pi\sqrt{d}}{\delta_T} \right)^{d-1} \log\left( \frac{2T^3\sqrt{d}}{\varepsilon_{\mathrm{hrd}}} \right),$$

*due to Lemma 3.4. Furthermore, for sufficiently large $T$, we have $\beta \leq d \cdot \Lambda$, where $\Lambda$ is the maximum number of refinements along any path in the hierarchical decomposition.*

*Proof.* By Lemma 3.4, the number of new regions added at any step is upper bounded by

$$\left( \frac{2\pi\sqrt{d}}{\delta_T} \right)^{d-1} \log\left( \frac{2T^3\sqrt{d}}{\varepsilon_{\mathrm{hrd}}} \right).$$

Each region $R \in \mathcal{R}_t$ can generate at most this many child regions $R' \subseteq R$ at the next level, which gives the stated bound on the maximum branching factor $\beta$.

Finally, since $\Lambda$ bounds the number of refinement levels along any path in the hierarchy, for sufficiently large $T$ the logarithmic term is dominated by $d \cdot \Lambda$, giving the inequality $\beta \leq d \cdot \Lambda$. $\qquad\square$

**Lemma 3.6.** *Consider an instance of the online low-rank approximation problem where a new unit vector $x_t \in \mathbb{R}^n$ arrives at each time step $t$, and let the spherical hierarchical region decomposition with parameter $\varepsilon_{\mathrm{hrd}}$ produce the regions $\mathcal{R}_t$ at step $t$. Let $\varepsilon_{\mathrm{hrd}} > 0$, and consider two multisets of rank-$k$ basis vectors $S = \{u_1, \ldots, u_k\}, S' = \{u'_1, \ldots, u'_k\}$, with $u_i, u'_i \in \mathbb{R}^n$ lying on the unit sphere for all $i \in [k]$.*

*Suppose that for each $i \in [k]$, the pair $(u_i, u'_i)$ lies within the same spherical region of the decomposition at step $t$. Then, for all $1 \leq \tau < t$,*

$$\ell(S', x_\tau) \leq (1 + \varepsilon_{\mathrm{hrd}})\, \ell(S, x_\tau) + \frac{\varepsilon_{\mathrm{hrd}}}{\tau^5}.$$

*In other words, for two points in the same region, the loss of $S'$ on any previously seen point is at most $(1 + \varepsilon_{\mathrm{hrd}})$ times the loss of $S$, up to a small additive term that decreases over time as $\frac{\varepsilon_{\mathrm{hrd}}}{\tau^5}$.*

*Proof.* Fix a previously seen point $x_\tau$. For each $i \in \{1, \ldots, k\}$, consider the optimal rank-$k$ vector $u_i \in U$. Let $R_i \in \mathcal{R}_t$ denote the region in the spherical hierarchical decomposition that contains $u_i$, and let $v_i$ be the centroid of $R_i$. By the construction of the spherical hierarchical decomposition:

(1) Each region $R_i$ is a subset of the unit sphere with angular diameter $\Delta_{R_i}$.

(2) The hierarchical decomposition guarantees that regions are refined whenever the angular distance from any point in the stream is large relative to the region size. Therefore, the diameter of $R_i$ is bounded by the *refinement criterion* applied at the time each point was inserted.

Formally, if $r_i := \mathrm{dist}_{\mathbb{S}^d}(x_\tau, u_i)$ denotes the geodesic (angular) distance between $x_\tau$ and $u_i$, then the refinement criterion ensures that either:

- the region is refined down to the minimum allowed angular diameter $\delta_\tau$ (corresponding to very small distances or late times), or

- the region's diameter is controlled proportionally to the distance $r_i$ as $\varepsilon_{\mathrm{hrd}} \cdot r_i/2$ (so that the local resolution adapts to the distance from the point).

Hence, for each $i$ we have

$$\mathrm{dist}_{\mathbb{S}^d}(u_i, v_i) \leq \Delta_{R_i} \leq \max\left( \varepsilon_{\mathrm{hrd}} \cdot r_i/2,\, \delta_\tau \right),$$

where $\Delta_{R_i}$ is the angular diameter of $R_i$.

Intuitively, this means that each centroid $v_i$ is guaranteed to be close to $u_i$ on the sphere, and the maximum distance is either determined by the minimum resolution of the grid or by a fraction of

the distance from $x_\tau$, whichever is larger. This bound will allow us to control the error introduced when replacing $u_i$ with $v_i$ in the low-rank approximation.

Let $\mathrm{proj}_U(x_\tau)$ denote the orthogonal projection of $x_\tau$ onto the subspace spanned by the optimal rank-$k$ vectors $U = \{u_1, \ldots, u_k\}$, and let $\mathrm{proj}_V(x_\tau)$ denote the orthogonal projection onto the subspace spanned by the centroids $V = \{v_1, \ldots, v_k\}$. We want to bound the difference in squared projection error:

$$\|x_\tau - \mathrm{proj}_V(x_\tau)\|_2^2 - \|x_\tau - \mathrm{proj}_U(x_\tau)\|_2^2.$$

Recall that each vector $v_i$ differs from $u_i$ by at most $\Delta_{R_i}$ along the surface of the unit sphere. Therefore, we can consider the effect of replacing $u_i$ with $v_i$ on the projection sequentially, one vector at a time. Using the triangle inequality, we have:

$$\|x_\tau - \mathrm{proj}_V(x_\tau)\|_2 \le \|x_\tau - \mathrm{proj}_U(x_\tau)\|_2 + \|\mathrm{proj}_U(x_\tau) - \mathrm{proj}_V(x_\tau)\|_2$$

where $U_{\backslash i}$ and $V_{\backslash i}$ denote the subspaces spanned by all vectors except the $i$-th. Squaring both sides gives

$$\|x_\tau - \mathrm{proj}_V(x_\tau)\|_2^2 \le \|x_\tau - \mathrm{proj}_U(x_\tau)\|_2^2 + 2\|x_\tau - \mathrm{proj}_U(x_\tau)\|_2 \cdot \|\mathrm{proj}_U(x_\tau) - \mathrm{proj}_V(x_\tau)\|_2$$
$$+ \|\mathrm{proj}_U(x_\tau) - \mathrm{proj}_V(x_\tau)\|_2^2.$$

The difference between the projections onto $U$ and $V$ can be bounded in terms of the deviations of the individual basis vectors:

$$\|\mathrm{proj}_U(x_\tau) - \mathrm{proj}_V(x_\tau)\|_2 \le \sum_{i=1}^{k} \|\mathrm{proj}_{u_i}(x_\tau) - \mathrm{proj}_{v_i}(x_\tau)\|_2,$$

where $\mathrm{proj}_{u_i}$ denotes projection onto the one-dimensional subspace spanned by $u_i$. Informally, we can use the small-angle approximation on the unit sphere, though extended to all ranges of $\theta$. Formally, for each $i$, we have

$$\|\mathrm{proj}_{u_i}(x_\tau) - \mathrm{proj}_{v_i}(x_\tau)\|_2 \le \frac{\pi}{2} \cdot \|x_\tau\|_2 \cdot \|u_i - v_i\|_2 \le \frac{\pi}{2} \cdot \Delta_{R_i}.$$

Let $r_i := \|x_\tau - \mathrm{proj}_{u_i}(x_\tau)\|_2$ denote the distance from $x_\tau$ to the component along $u_i$. Then, by the previous inequalities and the Cauchy–Schwarz bound for the cross term:

$$2\|x_\tau - \mathrm{proj}_U(x_\tau)\|_2 \cdot \|\mathrm{proj}_U(x_\tau) - \mathrm{proj}_V(x_\tau)\|_2 \le \pi \sum_{i=1}^{k} r_i \Delta_{R_i}.$$

Similarly, the squared perturbation term gives

$$\|\mathrm{proj}_U(x_\tau) - \mathrm{proj}_V(x_\tau)\|_2^2 \le \sum_{i=1}^{k} (\Delta_{R_i})^2.$$

Combining all of the above, we obtain

$$\|x_\tau - \mathrm{proj}_V(x_\tau)\|_2^2 \le \|x_\tau - \mathrm{proj}_U(x_\tau)\|_2^2 + \pi \sum_{i=1}^{k} r_i \Delta_{R_i} + \sum_{i=1}^{k} (\Delta_{R_i})^2.$$

Intuitively, the term $\pi r_i \Delta_{R_i}$ accounts for the linear change in distance caused by moving from $u_i$ to $v_i$, proportional to how far $x_\tau$ is from $u_i$, while the term $(\Delta_{R_i})^2$ accounts for the quadratic error introduced by the misalignment of the subspace.

The hierarchical region decomposition guarantees that each region $R_i$ satisfies the refinement criterion

$$\Delta_{R_i} \le \max\left(\frac{\varepsilon_{\mathrm{hrd}}}{2\tau^3}, \frac{\varepsilon_{\mathrm{hrd}} r_i}{2}\right).$$

We consider the two possible cases separately.

**Small-distance case:** $r_i \le 1/\tau^3$**.** In this case, the first term in the $\max$ dominates, so

$$\Delta_{R_i} \le \delta_\tau \le \frac{\varepsilon_{\mathrm{hrd}}}{2\tau^3}.$$

Then, the combined error term satisfies

$$r_i \Delta_{R_i} + \Delta_{R_i}^2 \leq r_i \frac{\varepsilon_{\mathrm{hrd}}}{2\tau^3} + \left(\frac{\varepsilon_{\mathrm{hrd}}}{2\tau^3}\right)^2 \leq \frac{1}{\tau^3} \cdot \frac{\varepsilon_{\mathrm{hrd}}}{2\tau^3} + \frac{\varepsilon_{\mathrm{hrd}}^2}{4\tau^6} \leq \frac{\varepsilon_{\mathrm{hrd}}}{\tau^5},$$

where the last inequality holds for sufficiently small $\varepsilon_{\mathrm{hrd}} \leq 1$ and $\tau \geq 1$.

**Large-distance case:** $r_i > 1/\tau^3$ Here, the second term in the $\max$ dominates, so

$$\Delta_{R_i} \leq \frac{\varepsilon_{\mathrm{hrd}} r_i}{2}.$$

Then the error term becomes

$$r_i \Delta_{R_i} + \Delta_{R_i}^2 \leq r_i \cdot \frac{\varepsilon_{\mathrm{hrd}} r_i}{2} + \left(\frac{\varepsilon_{\mathrm{hrd}} r_i}{2}\right)^2 = \frac{\varepsilon_{\mathrm{hrd}}}{2} r_i^2 + \frac{\varepsilon_{\mathrm{hrd}}^2}{4} r_i^2 \leq \varepsilon_{\mathrm{hrd}} r_i^2,$$

where the last inequality holds since $\varepsilon_{\mathrm{hrd}} \leq 1$.

By combining these two cases, we have a uniform bound on each term $r_i \Delta_{R_i} + \Delta_{R_i}^2$:

$$r_i \Delta_{R_i} + \Delta_{R_i}^2 \leq \max\left(\frac{\varepsilon_{\mathrm{hrd}}}{\tau^5}, \, \varepsilon_{\mathrm{hrd}} r_i^2\right),$$

which directly feeds into the bound on the squared projection error.

Finally, summing over $i = 1, \ldots, k$ and using $r_i^2 \leq \ell(U, x_\tau)$ for each component, we obtain

$$\ell(V, x_\tau) = \|x_\tau - \mathrm{proj}_V(x_\tau)\|_2^2 \leq (1 + \varepsilon_{\mathrm{hrd}})\ell(U, x_\tau) + \frac{\varepsilon_{\mathrm{hrd}}}{\tau^5}.$$

This proves that replacing the optimal basis vectors with the centroids of their spherical regions increases the loss by at most a multiplicative $(1 + \varepsilon_{\mathrm{hrd}})$ factor plus a small additive term that decays as $\tau^{-5}$. $\square$

### F.2 MTMW - MWUA FOR TREE STRUCTURED EXPERTS

**Lemma 3.9.** *Let $v$ be a region in the spherical hierarchical decomposition, $\tilde{\mathcal{T}}$ a subtree rooted at $v$, and $\tilde{V}$ the set of leaf regions of $\tilde{\mathcal{T}}$. Then the mass of $v$ equals the sum of the masses of its leaf regions, i.e., $M(v) = \sum_{v' \in \tilde{V}} M(v')$.*

*Proof.* We prove the statement by induction on the height $h$ of the subtree $\tilde{\mathcal{T}}$ rooted at $v$. We first consider the base case $h = 1$. If the subtree has height 1, then $\tilde{V} = \{v\}$, i.e., $v$ itself is a leaf. In this case,

$$\sum_{v' \in \tilde{V}} M(v') = M(v),$$

so the property holds.

For the inductive hypothesis, we assume the lemma holds for all subtrees of height $h$. Consider a subtree of height $h + 1$ rooted at $v$. Let $U$ denote the set of children of $v$, and let $\tilde{V}_{v'}$ denote the set of leaves of the subtree rooted at child $v' \in U$. By the inductive hypothesis,

$$\sum_{v'' \in \tilde{V}_{v'}} M(v'') = M(v').$$

Using the recursive definition of mass,

$$M(v') = \frac{M(v)}{|U|}.$$

Thus, summing over all children:

$$\sum_{v' \in \tilde{V}} M(v') = \sum_{v' \in U} \sum_{v'' \in \tilde{V}_{v'}} M(v'') = \sum_{v' \in U} M(v') = \sum_{v' \in U} \frac{M(v)}{|U|} = M(v),$$

where $\tilde{V}$ is the set of all leaves in the subtree rooted at $v$. This completes the induction, proving the lemma. $\square$

**Theorem 3.10.** *Let $\mathcal{T}_t$ be the tree corresponding to the spherical hierarchical region decomposition at step $t$. Consider running the Mass Tree Multiplicative Weights Update Algorithm (MTMW) over the set of experts given by all root-to-leaf paths of the final tree, $\mathcal{P}(\mathcal{T}_t)$. Even if the tree is revealed adaptively up to depth $t$ at each time step, running MTMW is possible provided each path $p \in \mathcal{P}(\mathcal{T}_t)$ is initialized with weight $M(p)$. Then, the regret of MTMW with respect to any path $p$ is bounded by*

$$Regret \leq \sqrt{-T \ln M(p)}.$$

*Moreover, the algorithm can be implemented with time complexity $O(|\mathcal{T}_t|)$, i.e., proportional to the number of vertices in the tree.*

*Proof.* Consider a rooted path $p = (v_1, \ldots, v_T)$ in the spherical HRD tree $\mathcal{T}_T$. At time $t$, define the cumulative weight of path $p$ as

$$u_p^{(t)} = \prod_{r=1}^{t-1} \left(1 - \eta \ell(v_r, x_r)\right),$$

where $\ell(v_r, x_r)$ is the loss of the low-rank approximation associated with vertex $v_r$ on the incoming unit vector $x_r$. This extends naturally to any path of length at least $t$.

For MTMW, the total weight of expert $p$ at step $t$ is

$$w_p^{(t)} = M(p)u_p^{(t)},$$

where $M(p)$ is the mass of the path as defined in Lemma 3.9. Let $p(v)$ denote the unique path from the root to vertex $v$. Then, the unnormalized probability that MTMW selects vertex $v_t$ at depth $t$ is

$$w_{v_t}^{(t)} = \sum_{p \ni v_t} w_p^{(t)} = \sum_{p \ni v_t} M(p)u_p^{(t)} = M(v_t)u_{p(v_t)}^{(t)},$$

where the final equality follows from the additive property of masses over descendant leaves, i.e., Lemma 3.9.

Analogously to the proof of Theorem (2.1) of Arora et al. (2012), we define the potential function at time $t$ as

$$\Phi^{(t)} = \sum_{p \in \mathcal{P}(\mathcal{T}_T)} w_p^{(t)}.$$

Let $m_p^{(t)} = \ell(p, x_t)$ denote the loss of path $p$ at step $t$, and let $p_p^{(t)}$ be the normalized probability of choosing expert $p$. Then, by the standard MWUA analysis, the potential at the final step satisfies

$$\frac{\Phi^{(T+1)}}{M(p)} \leq \frac{1}{M(p)} \exp\left(-\eta \sum_{t=1}^{T} m^{(t)} \cdot p^{(t)}\right).$$

On the other hand, the potential after the last round is at least the weight of path $p$ divided by its mass:

$$\frac{\Phi^{(T+1)}}{M(p)} \geq \frac{w_p^{(T+1)}}{M(p)} = u_p^{(T+1)}.$$

Combining these inequalities and following the standard MWUA regret analysis, we obtain that for any path $p \in \mathcal{P}(\mathcal{T}_T)$,

$$Regret \leq \sqrt{-T \ln M(p)},$$

where the inverse of the path mass captures the effective number of experts in the hierarchical decomposition.

Finally, the runtime of the algorithm is dominated by traversing the tree to update and normalize the weights along paths, which scales with the total number of vertices in the tree at the current depth. $\quad\square$

## F.3  REGRET BOUND

**Lemma 3.12.** *Let $\mathcal{H}$ be a spherical HRD with parameter $\varepsilon_{\mathrm{hrd}}$ that is constructed from $\{\mathcal{Q}_t\}_{t=1}^T$. Let $S^*$ be the best low-rank basis in hindsight, and let $\tilde{S}_t$ be the approximate basis induced by $\mathcal{H}$ at time $t$. Then, we have:*

$$\sum_{t=1}^T \ell(\tilde{S}_t, x_t) \le (1 + \varepsilon_c + 8(\varepsilon_{\mathrm{hrd}} + \varepsilon_c)k\Lambda)\,\mathcal{C} + k\Lambda$$

*where $C = \sum_{t=1}^T \ell(S^*, x_t)$ is the optimal offline reconstruction cost.*

*Proof.* Let $t_1, t_2, \ldots, t_{T_0}$ denote the time steps at which the approximate basis $\tilde{S}_t$ changes, i.e., $\tilde{S}_t \ne \tilde{S}_{t-1}$, with $t_1 = 1$ and $t_{T_0+1} = T + 1$. For all other $t$, we have $\tilde{S}_t = \tilde{S}_{t-1}$, so we can write

$$\sum_{t=1}^T \ell(\tilde{S}_t, x_t) = \sum_{i=1}^{T_0} \ell\left(\tilde{S}_{t_i}, X[t_i : t_{i+1} - 1]\right).$$

For each segment $[t_i, t_{i+1} - 1]$, we may bound the cumulative loss by considering the coreset approximation and the hierarchical refinement. Specifically, excluding the single-step refinements themselves and using that the loss is bounded by at most 1, since each vector is a unit vector, we have

$$\sum_{t=1}^T \ell(\tilde{S}_t, x_t) \le \sum_{i=1}^{T_0} \left( \ell\left(\tilde{S}_{t_{i+1}-1}, X[1 : t_{i+1} - 1]\right) - \ell\left(\tilde{S}_{t_{i+1}-1}, X[1 : t_i - 1]\right) \right) + T_0.$$

By the spherical coreset property (Lemma 3.9), for each segment we can relate the loss of the approximate basis to the optimal offline basis $S^*$:

$$\ell(\tilde{S}_{t_{i+1}-1}, X[t_i : t_{i+1} - 1]) \le (1 + \varepsilon_c)(1 + \varepsilon_{\mathrm{hrd}})\,\ell(S^*, X[t_i : t_{i+1} - 1]).$$

Combining over all segments, the sums telescope and we obtain

$$\sum_{t=1}^T \ell(\tilde{S}_t, x_t) \le \ell(S^*, X_{1:T}) + (\varepsilon_c + \varepsilon_{\mathrm{hrd}}) \sum_{i=1}^{T_0} \ell(S^*, X[t_i : t_{i+1} - 1]) + T_0.$$

Finally, applying Corollary 3.3 which bounds the total number of refinement steps by $T_0 \le k\Lambda$, and noting that $\sum_{i=1}^{T_0} \ell(S^*, X[t_i : t_{i+1} - 1]) \le \mathcal{C}$, we conclude

$$\sum_{t=1}^T \ell(\tilde{S}_t, x_t) \le (1 + \varepsilon_c + 8(\varepsilon_c + \varepsilon_{\mathrm{hrd}})k\Lambda)\,\mathcal{C} + k\Lambda,$$

as claimed. □

**Lemma 3.13.** *Let $p^*$ denote the path in the $k$-tree corresponding to the best set of basis vectors in hindsight. $\Lambda$ is the maximum refinement depth of the HRD and $\beta$ bounds the logarithm of the maximum branching factor of any region in the decomposition, as defined as in Corollary 3.3 and Corollary 3.5, respectively. Then we have $-\ln M(p^*) \le k^2 \Lambda \beta$.*

*Proof.* By Corollary 3.3 and the definition of $\Lambda$, the term $\log(\deg(v_t))$ can be nonzero at most $k\Lambda$ times. Each such occurrence is bounded by $k \cdot \beta$, since in the $k$-tree a node can be expanded into at most $((9\sqrt{d}/\varepsilon_{\mathrm{hrd}})^d \log(T^3))^k$ children. □

**Theorem 3.1.** *Algorithm 1 has a regret of $O\left( k \log\left( \frac{kT^3\sqrt{d}}{\varepsilon^2} \right) \sqrt{dT} \right) + \varepsilon\mathcal{C}$ and a runtime of $T \cdot \tilde{O}\left( \frac{\sqrt{d}k^2 T^3 \log T}{\varepsilon^2} \right)^{kd}$.*

*Proof.* First, we bound $\Lambda$. From the definition of the HRD refinement parameter,

$$\Lambda = \log\left(\tfrac{2T^3\sqrt{d}}{\varepsilon_{\mathrm{hrd}}}\right) \leq \log\left(\tfrac{4adT^6k^2}{\varepsilon^2}\right) \leq 2\log\left(\tfrac{akT^3\sqrt{d}}{\varepsilon^2}\right)$$

We now relate $\varepsilon_{\mathrm{hrd}}$ to $\varepsilon/(k\Lambda)$. By expansion:

$$\varepsilon_{\mathrm{hrd}} \cdot k\Lambda \leq \frac{\varepsilon}{2ak^2\log\left(T^3\sqrt{d}\right)} \cdot 2k\log\left(\frac{ak\sqrt{d}T^3}{\varepsilon^2}\right).$$

After simplifying logarithmic factors, this yields

$$\varepsilon_{\mathrm{hrd}} \cdot k\Lambda \leq 2\varepsilon\sqrt{a}.$$

By [Corollary 3.5](), for sufficiently large $T$ we have $\beta < d\Lambda$. Applying [Lemma 3.13](), MWUA achieves regret with respect to the best path bounded by

$$k\Lambda\sqrt{dT} \leq 2k\log\left(\tfrac{akT^3\sqrt{d}}{\varepsilon^2}\right)\sqrt{dT}.$$

For $a \geq 342$, we get $\sqrt{2\varepsilon/a} \leq \varepsilon/17$, which ensures that the bound of [Lemma 3.12]() controls the $\varepsilon$-approximate regret. Thus the final regret is

$$O\left(k\log\left(\frac{akT^3\sqrt{d}}{\varepsilon^2}\right)\sqrt{dT} + \varepsilon\mathcal{C}\right).$$

Additionally, since $|\mathcal{Q}_T| \leq O(\tfrac{k}{\varepsilon^2}\log^3 T)$, using [Lemma 3.4](), we can bound the $k$-tree vertices by the amount of leaves times the depth $T$; thus, we have:

$$\mathrm{Runtime} = O\left(T\left(|\mathcal{Q}_T|\left(\frac{2\pi\sqrt{d}}{\delta_T}\right)^{d-1}\log\left(\frac{2T^3\sqrt{d}}{\varepsilon_{\mathrm{hrd}}}\right)\right)^k\right)$$

$$= T \cdot \left(O\left(\frac{k}{\varepsilon^2}\log^3 T\right)\right)^k O\left(\left(\frac{\sqrt{d}}{\delta_T}\right)^{k(d-1)}\log^k\left(\frac{T\sqrt{d}}{\varepsilon_{\mathrm{hrd}}}\right)\right)$$

$$= T \cdot \left(O\left(\frac{k}{\varepsilon^2}\log^3 T\right)\right)^k O\left(\left(\frac{\sqrt{d}\,k^2 T^3\log T}{\varepsilon^2}\right)^{k(d-1)}\log^k\left(\frac{T\sqrt{d}\,k^2\log T}{\varepsilon^2}\right)\right)$$

$\square$

## F.4 Lower Bound

**Theorem D.2.** *Suppose there exists an online weighted low-rank approximation algorithm $\mathcal{A}$ that achieves regret $\tilde{O}(T^{1-\alpha})$ and, at time $t$, runs in time $f(t, k, d)$. Then, for any $\varepsilon > 0$, there exists a randomized offline algorithm that, given an instance of weighted low-rank approximation, outputs a solution with cost at most $C + \varepsilon$ with constant probability and runs in time polynomial in $n, k, d, \frac{1}{\varepsilon}$, and $f(n, k, d)$.*

*Proof.* We reduce an offline instance of weighted low-rank approximation to the online setting. Let $A \in \mathbb{R}^{n \times d}$ be the weighted input matrix, and suppose the goal is to find a rank-$k$ approximation minimizing the weighted Frobenius error. We construct a stream $X$ of $T$ rows, sampled uniformly with replacement from $A$, and feed these rows sequentially to the online algorithm $\mathcal{A}$. The algorithm produces a sequence of intermediate rank-$k$ projections $\{P_t\}_{t=1}^{T}$, and we output the projection with the smallest error when evaluated on the full dataset $A$.

Let $P^*$ denote the optimal offline rank-$k$ projection for $A$. The regret guarantee of $\mathcal{A}$ ensures that, if $R$ denotes the internal randomness of $\mathcal{A}$, then

$$\mathbb{E}_R\left[\mathbb{E}_X\left[\sum_{t=1}^{T}\ell(P_t, x_t) - \ell(P^*_{T+1}, X)\right]\right] \leq T^\alpha,$$

where $\ell(P, x)$ denotes the weighted squared error of approximating row $x$ by $Px$.

By linearity of expectation, and since $P^\star_{T+1}$ is independent of the randomness $r$, we obtain

$$\sum_{t=1}^{T}\mathbb{E}_R\left[\mathbb{E}_X\left[\ell(P_t, x_t)\right]\right] \leq T^\alpha + \mathbb{E}_X[\ell(P^*, X)].$$

For any projection $P$, the expected loss on a uniformly sampled row equals its normalized loss on the full matrix, i.e.

$$\mathbb{E}_{x_t}\left[\ell(P, x_t)\right] = \frac{1}{n}\,\ell(P, A).$$

Hence,

$$\sum_{t=1}^{T}\mathbb{E}_R\left[\frac{1}{n}\,\ell(P_t, A)\right] \leq T^\alpha + \frac{T}{n}\,\ell(P^*, A).$$

Define $\varepsilon_t \geq 0$ such that $\mathbb{E}_R\left[\ell(P_t, A)\right] = (1 + \varepsilon_t)\,\ell(P^*, A)$. Substituting gives

$$\sum_{t=1}^{T}(1 + \varepsilon_t)\frac{\ell(P^*, A)}{n} \leq T^\alpha + \frac{T}{n}\,\ell(P^*, A).$$

Rearranging yields

$$\sum_{t=1}^{T}\varepsilon_t \leq \frac{nT^\alpha}{\ell(P^*, A)}.$$

Let $\varepsilon^* = \min_t \varepsilon_t$. Then $T \cdot \varepsilon^* \leq \frac{nT^\alpha}{\ell(P^*,A)}$ implies $\varepsilon^* \leq \frac{nT^{\alpha-1}}{\ell(P^*,A)}$. Since $\ell(P^*, A) \geq \frac{1}{\text{poly}(n)}$ by assumption, choosing $T = \text{poly}(n)$ allows $\varepsilon^*$ to be made arbitrarily small. Thus, the online-to-offline reduction yields an efficient approximation scheme for weighted low-rank approximation, contradicting the known APX-hardness of WLRA (Gillis & Glineur, 2011), i.e., Theorem D.1. This completes the proof. $\square$

## G   OTHER EXPERIMENTS

### G.1   SYNTHETIC OPTIMAL DATASET AND FIXED BASELINE CONSTRUCTION

Our first synthetic dataset was constructed as a stream of points in the format $(x_1, x_2, 0, 0, 0)$. The baseline was constructed as $k$ fixed basis vectors in $d$-dimensional space using a deterministic pattern. For each basis vector $v_i$, the $j$-th coordinate is generated as $\text{coordinate}_j = \text{sign}_j \times \frac{j+1}{3}$, where $\text{sign}[j]$ is a randomly assigned $\pm 1$ value that remains fixed for the duration of the experiment. Each resulting vector is then normalized.

**Experimental setup.** For the purpose of reproducibility, our experiments were conducted with Python 3.13.7 on a 64-bit operating system with an Intel Core i7-1165G7 CPU with 16 GB DDR4 RAM and 4 cores with base clock speed 2.70 GHz. The following core python packages are required: NumPy 2.2.6 for numerical computations and array operations, Matplotlib 3.10.6 for data visualization and plot generation, scikit-learn 1.7.2 for dimensionality reduction (TruncatedSVD) and machine learning utilities, and Pandas 2.3.2 for data manipulation and CSV file handling.

In these experiments, we compared the performance of low-rank approximation using our spherical partitioning algorithm with the performance of low-rank approximation using the fixed baseline net. We set the basis dimension to test at $k = 2$, $k = 5$, and $k = 10$ to provide a manageable level of complexity while testing whether increasing the dimension would impact its effectiveness, and streamed 1000 data points. The full vector dimension is set to $d = 5$. The HRD algorithm

uses splitting threshold $d_{split} = 15$ (75% of ambient dimension) to prevent premature splits in high-dimensional regions while allowing refinement when sufficient data concentration occurs. Minimum leaf size $n_{min}$ was set to 20, and maximum leaf size $n_{max}$ to 100. The minimum prevents splits with insufficient statistical support, while the maximum forces splits when nodes become overcrowded. The learning rate is $\eta = 0.5$, which we have kept moderate so that adaptation is not overly sensitive and also not too conservative. The refinement parameter $\varepsilon_{hrd}$ is set to 0.1, another moderate choice to prevent unnecessary splits that do not add much to approximation accuracy while also not preventing actually beneficial splits. The reconstruction error is calculated as the Frobenius norm reconstruction error, and the total cumulative loss is the loss accumulated starting from time step 0.

Moreover, in our experiments, we chose to split spherical regions using up to $d$ orthogonal hyperplanes instead of splitting via angular coordinate bisection along polar coordinates as described in Section 3.2. For this process, we first attempt to use directions that were unused in previous splits by projecting them orthogonally to the centroid $c$ of the current node. Then, when insufficient reusable directions exist, we generate random vectors orthogonal to $c$ and orthogonalize them against existing directions using the Gram-Schmidt process. By using these random orthogonal vectors for splitting, we get a more practical implementation that uses standard linear algebra operations rather than complex angular coordinate manipulations, while achieving approximately equivalent region refinements and maintaining the fundamental guarantee that region diameters decay exponentially with tree depth.

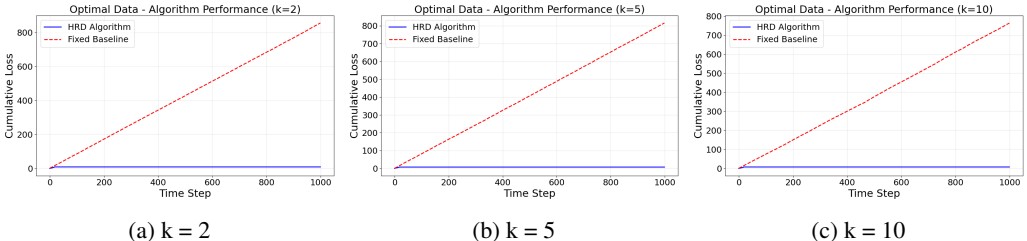

(a) k = 2          (b) k = 5          (c) k = 10

Fig. 2: Cumulative loss comparisons of HRD Algorithm and Fixed Baseline Net for optimal data with different values of k

**Results and discussion.** Our observations in Figure 2 on the optimal synthetic dataset show that the HRD algorithm significantly outperforms the baseline net, with final cumulative loss of 8.7454 for the HRD algorithm and 856.0313 for the baseline net given basis dimension k = 2. This is an improvement of around 98.98%. The HRD algorithm demonstrates adaptive learning as time increases, successfully identifying and exploiting the optimal 2-dimension subspace within the 5-dimension ambient space. The cumulative loss stays relatively constant throughout the whole time sequence after an initial small increase. Meanwhile, the cumulative loss for the baseline non-adaptive net increases linearly, leading to the difference in performance becoming increasingly pronounced as more data points appear. Moreover, for the k = 5 and k = 10 cases, we see similarly large gaps in performance. This demonstrates that when data lies in a low-dimensional subspace, HRD can discover and exploit this structure.

G.2   SYNTHETIC DATASET WITH CLUSTERS

To evaluate the performance on synthetic data with natural groupings, we performed the experiment again using synthetic data with three major clusters.

**Experimental setup.** We created three cluster centers by sampling random $d$-dimensional vectors from a standard Gaussian distribution and normalizing them to unit length. The process is as follows: for each of the 1000 data points, we select a cluster center in round-robin fashion. Each point is then generated by adding Gaussian noise with standard deviation $\sigma = 0.3$ to the selected cluster center and then normalizing the resulting vector to unit length to maintain spherical constraints. Each cluster center represents a different region on the unit sphere. We then tested the HRD algorithm against the non-adaptive baseline for target basis dimension k = 2, k = 5, and k = 10. The other parameters remain the same as in the optimal dataset experiment.

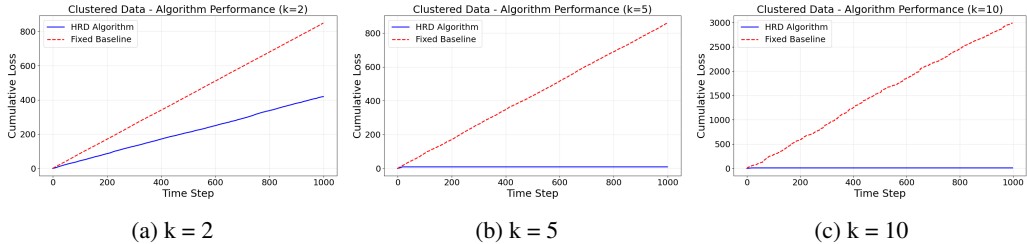

(a) k = 2        (b) k = 5        (c) k = 10

Fig. 3: Cumulative Loss of HRD Algorithm and Fixed Baseline Net over 1000 Data Points for Clustered Data with different values of k

**Results and discussion.** Our observations in Figure 3 on the synthetic dataset with clusters show that the HRD algorithm still outperforms the baseline net, even though the rate of increase for both is linear. The final reconstruction loss for the HRD algorithm is $419.3647$, while the loss for the baseline is $848.8485$, giving us an improvement of $50.60\%$ for the k = 2 case. This indicates that the algorithm is able to identify the underlying three-cluster structure, adaptively partitioning the spherical space to capture the distinct groupings.

Next, when we increase the rank of our basis vectors up to 5 and then 10, the HRD algorithms' performance improves even more significantly, with reconstruction loss staying flat at the bottom of the graph while the loss of the fixed baseline continues to rise linearly as before. The algorithm gives us improvements of $98.94\%$ and $99.64\%$ for k = 5 and k = 10, respectively. These results demonstrate that with higher rank basis vectors, HRD can better capture the underlying cluster structure through adaptive partitioning.

## G.3   CREDIT CARD DATASET

We also tested on a second real-world dataset, the Credit Card Fraud Detection dataset Dal Pozzolo et al. (2014), which has 31 features and 284,807 observations. This dataset contains credit card transactions made in two days in 2013, with 492 of them being fraudulent transactions. The features are numerical input variables which are the result of transformation via PCA. This dataset was accessed through Kaggle.

**Experimental setup.** To process the data, we first removed three non-essential columns from the dataset (Time, Amount, Class). The Time feature does not provide any structural information for the low-rank approximation. The Class feature is the fraud label and should not be included in the feature representation. The Amount feature, while potentially informative, has a different scale than the PCA-transformed V1-V28 features and could dominate the approximation. This preprocessing yields 28-dimensional feature vectors corresponding to the principal components V1-V28. Next, all vectors are normalized to ensure that they lie on the unit sphere. Next, we tested on target basis vector dimension k = 10, 15, and 20 to analyze the effects of increasing dimensionality on the performance. The splitting dimension $d_{split}$ is set to 15, as in the synthetic experiments. Due to the large dataset size, we process the first 500 points, as in the MNIST experiment. We then run the HRD algorithm against the fixed baseline described in the synthetic experiments section.

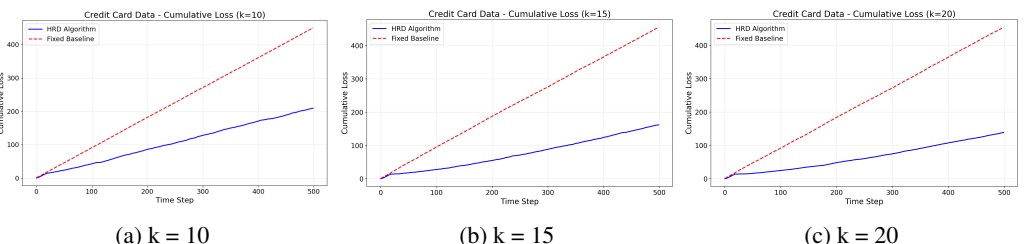

(a) k = 10        (b) k = 15        (c) k = 20

Fig. 4: Cumulative Loss of HRD Algorithm and Fixed Baseline Net over 500 Data Points for Credit Card Fraud Dataset with different values of k

**Results and discussion.** The results in Figure 4 show that the HRD algorithm outperforms the fixed baseline over the credit card dataset, as it consistently achieves lower cumulative loss at each time step, with total improvements of $53.25\%$, $53.25\%$, and $64.45\%$ for basis vector dimension k = 10, 15, and 20 respectively. These results indicate that the HRD algorithm is able to take advantage of the structural patterns in the credit card dataset. Notably, on this credit card dataset, performance isn't improved as significantly as in the MNIST dataset, indicating that the PCA-transformed features may not exhibit structure that is as easily exploitable by low rank approximation.

