# OpenReview forum: "Online Low-Rank Approximation via Adaptive Spherical Partitioning"
_ICLR.cc/2026/Conference — ICLR 2026 Conference Desk Rejected Submission_

### Official Review · Reviewer_oGcz · 2025-10-22

**Soundness:** 2
**Presentation:** 2
**Contribution:** 2
**Rating:** 4
**Confidence:** 3

**Summary:**

This paper studies the online low-rank approximation problem, where a sequence of points the goal is to find the best rank-$k$ subspace in an online fashion. The algorithm combines the spherical hierarchical region decomposition and the MWU framework to achieve sublinear regret. The main idea is to construct an adaptive coreset. The major issue is that the algorithm runs in exponential time. While the authors justify it by presenting a reduction from offline weighted low-rank approximation to online weighted low-rank approximation, unweighted low-rank approximation is known to be solved in polynomial, even input sparsity time. Experiments are performed on both synthetic datasets (appendix) and MNIST to show that the proposed algorithm achieves lower loss than the baseline that picks a fixed rank-$k$ basis.

**Strengths:**

1. The problem being studied is quite important. Given a sequence of vectors arrived in an online fashion, finding the best rank-$k$ subspace to approximate them holds both theoretical and practical impacts.

2. The overall presentation is quite clear.

**Weaknesses:**

1. The spherical HRD + MWU framework has been studied in [1] for the $k$-means problem. The overall algorithm, many parts of the analysis and even the experiments are very similar to that of [1]. I would argue the structure of the two papers are similar, e.g., compare Section 2 of this paper with Section 1.1 of [1]. However, authors only sparsely mentioned [1], making it hard to judge the novelty of the paper. It seems to me that the main difference is that [1] uses axis-aligned grid over $[0, 1]^d$, while this paper uses spherical HRD on $\mathbb{S}^{d-1}$? Also, in Section 3.3, authors claimed that they combine the spherical HRD of [1] and the MWU regret bound of [2], however, it seems that in [1], these two notions have already been combined and analyzed. I believe it is crucial for authors to properly cite which parts are inspired or are variants of [1], and which parts need to be adapted for low-rank approximation.

2. The exponential runtime is hard to justify. While the authors argue that one can convert an online weighted low-rank approximation algorithm to an offline one, and the NP hardness [3] or even SETH hardness [4] to approximate kicks in, for standard low-rank approximation with $W$ being the all-1's matrix, polynomial time and even input sparsity time algorithms are known. Meanwhile, the proposed algorithm in this paper does not differentiate between these two cases at all. Also, I think the wording of the ``Hardness results and offline-to-online connections" could be made more clear, i.e., the reduction is between weighted low-rank approximation (with the same weights), not that an online unweighted low-rank approximation can be reduced to an offline weighted low-rank approximation.

References:

[1] Cohen-Addad, Guedj, Kanade and Rom. Online $k$-means Clustering.

[2] Arora, Hazan and Kale. The multiplicative weights update method: a meta algorithm and applications.

[3] Gillis and Glineur. Low-rank matrix approximation with weights or missing data is np-hard.

[4] Razenshteyn, Song and Woodruff. Weighted Low Rank Approximations with Provable Guarantees.

**Questions:**

Comments and questions:

* The $d$-dimensional unit sphere is usually denoted as $\mathbb{S}^{d-1}$ rather than $\mathbb{S}^d$. Also, there are many mix uses of $\mathbb{S}^d$ and $S^d$. Authors should stick to one consistent notation.

* Line 208, if q(R) then -> if $q(R)$ then.

* Line 223, $d-dimensional$ -> $d$-dimensional.

* Line 234, $R_{\in} \mathcal{R}_\tau$ -> $R\in \mathcal{R}\_{\tau}$.

* Line 257, should $x_d=\sin(\theta_1)\ldots\sin(\theta_{d-2})\cos(\theta_{d-1})$?

* For the matrix $A$, it is sometimes denoted as $\mathbf{A}$ or $A$. Notation should be consistent.

* What is the baseline algorithm on synthetic datasets? It seems that authors move the synthetic experiments to the appendix, but didn't move the discussion on the baseline algorithm back to the main body of the work. Also, the baseline is a bit too weak.

* The use of $k$ and k is inconsistent.

* Line 687, "multiplicative weighs" -> "multiplicative weights".

* [4] provides more fine-grained complexity analysis for weighted low-rank approximation. It also gives several scenarios where the problem can be solved in polynomial time.

References:

[4] Razenshteyn, Song and Woodruff. Weighted Low Rank Approximations with Provable Guarantees.

---

### Official Review · Reviewer_rrVo · 2025-10-24

**Soundness:** 2
**Presentation:** 1
**Contribution:** 3
**Rating:** 4
**Confidence:** 2

**Summary:**

The submission studies Online Low-Rank Approximation and provides a novel algorithm which achieves an essentially square-root regret bound. It also provides lower bounds and an empirical evaluation.

**Strengths:**

The proofs are non-trivial and the contributions include a nice combination of foundational and empirical results.

**Weaknesses:**

-Has the online formulation of low-rank approximation been studied before? The only citation provided seems to be for

"Michael Kamp and Mario Boley. Streaming fraud detection: A survey and new research directions. Data Mining and Knowledge Discovery, 33(2):497–531, 2019."

which, however, does not seem to exist. If that is indeed the case then one is left wondering: why does the submission cite a non-existent article as its sole reference for its primary problem of interest (i.e., online low-rank approximation)?

Essentially, the article treats the problem and related notions as well-established, but does not provide any references or explanations substantiating this. This is a critical flaw: without convincing connections of the studied problem to the core topics of the conference, the article would not be of interest to the ICLR community (the employed techniques, while non-trivial, are mostly of interest to TCS researchers). This issue also arises in the empirical evaluation at the end of the paper: the submission claims that its implementation "consistently outperforms standard baselines", but never explains what these baselines are, and I could not find any references to these.

Further, the main contributions of the article are not explained very well. The first two pages of the article does not discuss the contributions at all, and even the "Our Contributions" section starts with a lengthy discussion of the generalizability of the supposed results - even though the actual results are not even stated at that point.

**Questions:**

The algorithm listed as the first contribution provides a square-root regret bound. At the same time, the second contribution (the lower bound) essentially excludes achieving sublinear regret in polynomial time. Do I understand correctly that these two results can coexist because the former algorithm does not run in polynomial time?

Aside from answering the above question, the authors are of course also welcome to respond to the above-mentioned weaknesses.

---

### Official Review · Reviewer_EyWZ · 2025-11-06

**Soundness:** 2
**Presentation:** 2
**Contribution:** 1
**Rating:** 2
**Confidence:** 3

**Summary:**

This paper is about the online SVD problem, where the algorithm is required to maintain a low-rank projection $\Pi_t$ over time $t$ to minimize $\sum_t ||(I-\Pi_t) x_t||^2$ for an online sequence of vectors $x_t$. The authors follow a regret minimization approach based on the multiplicative weights update algorithm over unit vectors, and show that sublinear regret can be achieved if we allow an additional (1+eps) multiplicative regret factor (*). The approach and analysis is based on that of Cohen-Addad et al (AISTATS 2021) for the online k-means problem. The authors also show that the extra multiplicative term (*) is necessary for polynomial-time sublinear regret algorithms, under some APX-hardness assumption. Some experimental results are also presented that compare thee uathors' algorithm with a baseline eps-net.

**Strengths:**

* The parts of the analysis that I checked look sound.

* The online SVD problem has a lot of applications.

**Weaknesses:**

* The main result has very high computational complexity, roughly $T^{O(kd)}$, where $T$ is the number of rounds, $k$ is the low-rank dimension, and $d$ is the ambient dimension. In contrast, in the result of Cohen-Addad et al the complexity is much lower, i.e. $(\log T)^{O(kd)}$. In fact, the $T^{O(kd)}$ bound should be achievable by an arbitrary $\epsilon$-net (this is also done in Cohen-Addad et al. Theorem 2.1, and the paper is devoted to reducing the polynomial runtime dependency on $T$), without the need to construct adaptive nets. I believe this makes the contribution of hte paper marginal.

* There is extensive previous work on the online svd problem, none of which is cited. See e.g. https://jmlr.org/papers/volume17/15-320/15-320.pdf https://proceedings.mlr.press/v70/allen-zhu17d/allen-zhu17d.pdf https://edoliberty.github.io/papers/opca.pdf. The authors should contextualize their approach in these previous works.

* The authors' approach very closely follows Cohen-Addad et al. However it is not clear which parts are novel and which parts are from Cohen-Addad et al. The authors should more precisely outline their contribution.

* The experimental baseline is quite weak.

**Questions:**

How does the runtime of the authors' approach compare to MWU on a fixed eps-net?

---

### Note · Program_Chairs · 2026-01-17
**Submission Desk Rejected by Program Chairs**

The following references in this submission do not refer to real documents and/or have major errors in bibliographic information:

 Michael Kamp and Mario Boley. Streaming fraud detection: A survey and new research directions. Data Mining and Knowledge Discovery, 33(2):497-531, 2019. I